# Organic iron-binding ligands mediate dissolved-particulate exchange in hydrothermal vent plumes along the mid-Atlantic Ridge

Travis Mellett[1,2], Justine B. Albers[3], Alyson E. Santoro[3], Pascal Salaun[4], Joseph Resing[5,6], Wenhao Wang[7], Alastair J.M. Lough[7,8], Alessandro Tagliabue[4], Maeve Lohan[7], Randelle M. Bundy[2], Kristen N. Buck[1,9].

[1]University of South Florida, College of Marine Science, St. Petersburg, FL, USA
[2]University of Washington, School of Oceanography, Seattle, WA, USA
[3]University of California, Department of Ecology, Evolution and Marine Biology, Santa Barbara, CA, USA
[4]University of Liverpool, School of Environmental Sciences, Liverpool, UK
[5]NOAA Pacific Marine Environmental Laboratory, Seattle, WA
[6]University of Washington's Cooperative Institute for Climate, Ocean, and Ecosystem Studies, Seattle, WA, USA
[7]School of Ocean and Earth Science, University of Southampton, National Oceanography Centre Southampton, Southampton, UK
[8]University of Leeds, School of Geography, Leeds, UK
[9]Oregon State University, College of Earth, Ocean, and Atmospheric Sciences, Corvallis, OR, USA

*Corresponding author:* Travis Mellett (tmellett@uw.edu)

*Abstract*

Hydrothermal vents are important contributors to the dissolved iron inventory in the ocean. Investigating the processes underlying iron behavior in hydrothermal plumes is challenging, but important for constraining deep ocean iron cycling. Field studies suggest that the retention of hydrothermal iron in the deep ocean is primarily supported by two mechanisms: the formation of colloidal nanoparticles and the stabilization of iron by organic ligands. Here we present a novel dataset from shipboard incubation experiments designed to investigate the interplay between these two processes and how they contribute to the stabilization of iron away from ridge axes. Filtered and unfiltered water collected from the hydrothermal plumes of three vent fields along the Mid-Atlantic Ridge as part of GEOTRACES cruise GA13 was incubated in the dark and regularly sampled over time (up to 3 weeks) for concentrations of size-fractionated iron and iron-binding ligands, for dissolved iron isotopic composition, and for microbial community composition. We observed rapid exchange of iron between physicochemical phases that appeared to be mediated in part by organic iron-binding ligands at each stage of plume evolution. Weaker iron-binding ligands sources from the vents were largely lost to the particulate phase with colloidal Fe phases via aggregation early in plume development, similar to the loss of iron and organic matter commonly observed in estuarine systems. Soluble organic ligand production was observed in later stages of all unfiltered incubations followed by mobilization of particulate and colloidal Fe into the soluble phase in the longer incubations, revealing a potentially important mechanism for generating the persistent iron observed in long-range plumes.

*1.0 Introduction*

Iron (Fe) is a globally important micronutrient in marine biogeochemical systems, limiting primary production and the microbial loop over large areas of the surface ocean (Martin and Fitzwater, 1988; Moore et al., 2013, 2001; Li et al., 2024; Manck et al., 2024). Delivery of Fe to the remote surface ocean is achieved through atmospheric transport of dust aerosols (Johnson et al., 1997), but also via long-range transport and subsequent upwelling of subsurface Fe sources from the continental margin (Lam et al., 2006) and from hydrothermal vents (Tagliabue et al., 2014; Resing et al., 2015). Hydrothermal vents in particular, are a more recently recognized important contributor to the overall dissolved Fe inventory in seawater, with observations from the GEOTRACES program reporting long-distance dispersion of vent sourced Fe; in some cases, hundreds to thousands of kilometers away from the ridge axis (Wu et al., 2011; Klunder et al., 2011; Kondo et al., 2012; Nishioka et al., 2013; Resing et al., 2015; Fitzsimmons et al., 2017), and have been estimated to contribute up to 9% of the Fe in the deep ocean (Sander and Koschinsky 2011). Depending on the location and longevity of the Fe from the vent source, it could also influence surface Fe biogeochemistry and impact primary productivity and carbon export (Resing et al. 2015). However, hydrothermal systems are highly variable and notoriously difficult to study, so quantifying the global impact of Fe sourced from vents remains an important research question.

Hydrothermal plumes represent a stark physical and chemical boundary system in the deep ocean water column. When hot (~350 °C), acidic, and reduced hydrothermal fluids are expelled at the base of a cool (~2–4 °C), stratified water-column, a buoyant plume is formed. As the plume mixes with surrounding deep ocean waters it eventually becomes a neutrally buoyant plume (Lupton et al., 1985) and may be transported laterally along isopycnals from the vent site via local currents. During this transition from a buoyant to neutrally buoyant plume, the majority of Fe (40–90%) present in the vent fluid is lost to precipitation as large sulfide and Fe-(oxyhydr)oxide particles (Campbell et al., 1988; German et al., 1991; Mottl and McConachy, 1990; Rudnicki and Elderfield, 1993; Lough et al., 2019), and any remaining Fe can be transported away from the vent site.

In oxygenated seawater, the solubility of inorganic Fe is exceedingly low (Liu and Millero, 2002) and should preclude long-range transport of Fe from vent systems. Field studies have identified two primary mechanisms that can support the observed far-field dispersion of Fe: (1) the formation of small inorganic nanoparticulate pyrites that are resistant to oxidation and sink more slowly than larger particulate forms (Hochella et al., 2008; Yücel et al., 2011; Gartman et al., 2014; Gartman and Luther, 2014), and (2) the stabilization of dissolved Fe by Fe-binding organic ligands. Several studies have also shown that hydrothermal vents are a source of natural Fe-binding ligands above background deep ocean concentrations (Bennett et al., 2008; Sander and Koschinsky, 2011; Hawkes et al., 2013b; Kleint et al., 2016) and the upregulation of siderophore uptake and biosynthesis genes within a hydrothermal plume suggests this organic complexation may stabilize Fe in the dissolved phase (Li et al., 2014). These two mechanisms are not mutually exclusive, and their interplay may explain the low-density organic matrices of particles observed far-field of vent systems (Fitzsimmons et al., 2017; Hoffman et al., 2020).

Thus, there is compelling evidence from previous studies for both inorganic and organic mechanisms for the stabilization of Fe emitted from hydrothermal vents, with both likely playing an important role in the evolution of dissolved Fe in hydrothermal plumes. However, most field studies are limited to sampling vent plumes at a single snapshot in time, limiting our understanding of how inorganic nanoparticles and/or organic ligands contribute to stabilizing Fe over time during the different physical (buoyant/neutrally buoyant) and chemical (reducing/oxidized) stages of

plume evolution and advection far-field. To date, only one study has attempted to observe the temporal evolution of dissolved Fe in plume waters through a 24 hour incubation of a Niskin bottle (Lough et al., 2017).

Here we present the results from four incubation experiments aimed at examining processes underpinning Fe cycling during hydrothermal plume formation and evolution. Plume water was incubated filtered (< 0.2 µm) and unfiltered in the dark for timescales ranging from 6-22 days. Experiments were sampled for three different physicochemical size-fractions (dissolved, soluble, total particulate) for Fe concentrations, Fe-binding organic ligands, dissolved Fe isotopes, and 16S microbial community composition. The results from these experiments revealed how both newly produced and ambient organic ligands are involved in the exchange of Fe between dissolved and particulate phases in plume waters over different timescales. Together, these data provide novel insights into the mechanisms that support long-range dispersal of hydrothermal Fe in the deep sea.

## *2.0 Methods*
### *2.1 Incubation locations, setup, and sampling*

Hydrothermal plumes from three vent systems (Lucky Strike, Rainbow, and TAG) along the MAR were sampled on the *RSS James Cook* (JC156, UK GEOTRACES voyage GA13) between 20 December 2017 and 1 February 2018 (Figure 1) using a titanium frame 24-bottle rosette (SeaBird Scientific) equipped with trace metal clean Teflon-coated 10 L x-Niskin samplers (Ocean Test Equipment, Inc.). Hydrothermal plumes were identified using a CTD-mounted light scatter sensor (LSS) as an optical proxy for particles and an oxidation-reduction potential (ORP) sensor for detection of reduced species contributed from vent fluid end members. Once on board, the x-Niskins were drained into acid-cleaned, Milli-Q (MQ) conditioned, and seawater-rinsed 20 L polycarbonate (PC) carboys in a class-1000 clean air van. Each incubation carboy was filled from two x-Niskin bottles, closed in succession during the upcast of the plume sampling casts. In addition to unfiltered water for the incubations, we also incorporated filtered (<0.2 µm, Acropak) plume waters as additional treatments for the TAG and Rainbow near-field incubations (Figure 1). Thus, the filtered treatment represents the water that was filtered prior to the start of the incubation, whereas unfiltered treatment refers to incubations of hydrothermal plume water still containing ambient particles and microbial communities. All incubation carboys were kept in the dark in a controlled temperature room (14 ± 1 ºC) aboard the ship.

Subsampling of the incubation carboys was conducted in a laminar flow clean hood. Each incubation carboy was gently inverted at least 3 times to homogenize any settled particles before distributing an aliquot into 2.5 L PC bottles that were cleaned, conditioned, and sample rinsed prior to filling. At the time of subsampling, total dissolvable ("TD") trace metal samples were collected directly from the carboy. Contents of the 2.5 L bulk aliquot for were sequentially filtered through 3 µm and 0.4 µm acid-cleaned polycarbonate track etched (PCTE) filters in 47 mm Teflon filter holders (Savillex) on a two-stage custom-built filtration rig for dissolved ("d") trace metal and Fe speciation samples. The remaining unfiltered subsample was filtered through a 0.02 µm Anodisc filter to collect soluble ("s") samples. For selected incubations (Rainbow near-field, Rainbow far-field, Lucky Strike), microbial community subsamples were taken at the start of the incubation and again after one week (Rainbow near-field) or 2 weeks into the incubation for longer experiments (Rainbow far-field, Lucky Strike).

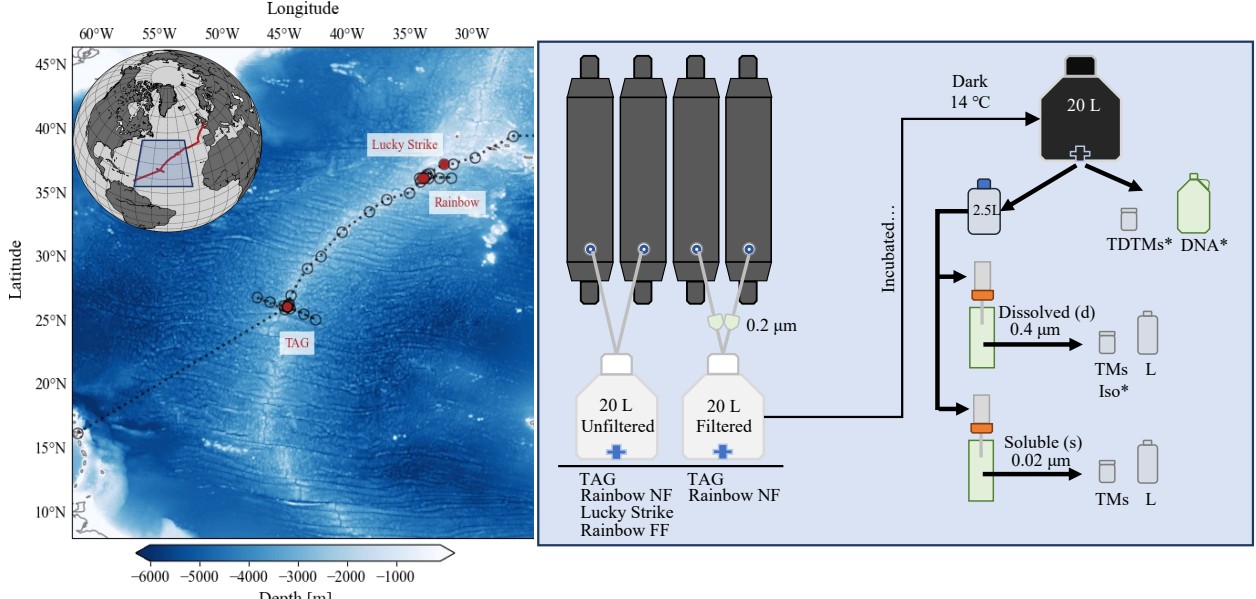

Figure 1. Overview map and flowchart of incubation setup. Left panel: Map of GA13 cruise stations; red circles denote
locations of incubation water collection (Lucky Strike, Rainbow, and TAG). Right panel: Flowchart of the incubation
setup and sampling. Below the incubation carboys is a list of vent sites the treatment was applied (TAG, Lucky Strike,
Rainbow near-field (NF) and far-field (FF)). The samples taken include total dissolvable trace metals (TDTMs), Fe
isotopes (Iso), 16S (DNA), dissolved (d) and soluble (s) trace metals (TMs) and Fe speciation (L) described in further
detail in section 2.1. *Indicates samples that were not sampled each time point.

### 2.2 Quantification of total dissolvable (unfiltered), dissolved (<0.4 μm) and soluble (<0.02 μm) trace metals

All samples for trace metal concentration analysis were collected in 125 mL low-density polyethylene (LDPE; Nalgene) bottles and acidified with ultrapure hydrochloric acid (HCl, Romil-UpA; final 0.024 M HCl, pH ~ 1.7–1.8) and stored for at least 6 months prior to analysis. Concentrations of both total dissolvable and dissolved Fe and Mn in all samples were determined first by direct injection high resolution inductively coupled plasma-mass spectrometry (HR-ICP-MS; Thermo Scientific Element XR) at the University of South Florida after diluting 50-fold with 1 N quartz-distilled nitric acid ($HNO_3$). Samples with concentrations <25 nM Fe required preconcentration for accurate quantification. A seaFAST-pico system was used to buffer the samples to pH $6.2 \pm 0.2$ with a 5.4 M ammonium acetate buffer before preconcentration of the trace metals in the samples on a Nobias PA1 resin. The preconcentrated sample was then eluted from the resin with 1 N quartz-distilled nitric acid, and the concentrations of Fe, Mn, and vanadium (V) analyzed on an Element XR HR-ICP-MS (Hollister et al., 2020). Quantification of trace metals by standard addition following normalization of counts to internal standards of indium and rhodium were used for all HR-ICP-MS analyses. Accuracy was verified by analysis of the certified reference materials CASS-6, NASS-7 (National Research Council of Canada), and SAFe D2 (Johnson et al., 2007). A comparison of our results with consensus values are presented in the supporting information (Table S1).

### 2.3 Fe isotope measurements

For the two high-Fe incubations conducted at Rainbow and TAG, the dissolved trace metal samples were subsampled and analyzed for dissolved Fe isotopes ($\delta^{56}dFe$) following the methods described in Wang et al., (2021) using distilled UpA grade reagents at the National Oceanography

Centre in Southampton UK. The isotopic composition of Fe was determined by multicollector
inductively coupled plasma mass spectrometry (MC-ICP-MS; Thermo Fisher Neptune Plus) at the
University of Southampton. The raw data were corrected for instrumental mass bias using an
iterative deconvolving procedure (Albarede and Beard, 2004). The final Fe isotope value of the
samples is reported in delta notation relative to the IRMM Fe isotope standard and expressed as:
$$\delta^{56}Fe \ (‰) = [(^{56}Fe/^{54}Fe)_{sample}/(^{56}Fe/^{54}Fe)_{IRMM-14} - 1] \times 1000 \tag{1}$$
Long-term analyses of the ETH Fe isotope standard gave $\delta^{56}Fe = 0.51 \pm 0.09‰$ (2SD, $n$=45), in
agreement with the consensus value (+0.52 ± 0.08‰; Lacan et al., (2010)). The accuracy of the
method was further validated through the analysis of trace metal free seawater doped with the
hematite Fe isotope standard, yielding an average $\delta^{56}Fe$ value of +0.22 ± 0.10‰ (2SD, $n$=5),
consistent with previously published hematite values ($\delta^{56}Fe$ = +0.24 ± 0.05‰; (Klar et al., 2017).

### 2.4 Fe-binding ligands by forward and reverse titration

Filtered (< 0.2 μm, <0.02 μm) samples were collected for Fe-binding organic ligands ($L_{Fe}$)
in acid-cleaned, MQ conditioned, and sample-rinsed 500 mL fluorinated polyethylene (FPE)
bottles (Nalgene) and stored frozen (-20 °C) until analysis at the University of South Florida. Two
different methods of competitive ligand exchange-adsorptive cathodic stripping voltammetry
(CLE-AdCSV) were employed for measuring Fe-binding organic ligands in our samples: forward
titrations (Rue and Bruland, 1995; Buck et al., 2018; Mahieu et al., 2024) and reverse titrations
(Hawkes et al., 2013a). When dFe concentrations exceeded ligand concentrations, we used
'reverse titrations' to allow quantification of Fe-binding ligands bound to ambient Fe by
progressively increasing the concentration of an electroactive ligand until all exchangeable Fe is
removed from the natural ligands (Hawkes et al., 2013a). We applied both methods for the
Rainbow far-field and Lucky Strike incubations since we were unsure of the extent of ligand
saturation in the samples prior to analyses. We report forward titration results for samples with
excess ligands observed in the titrations, and results from reverse titrations when initial titration
points in the forward titrations indicated ligands were already saturated with Fe.
For the forward titrations, we used the competitive ligand salicylaldoxime (SA; Buck et
al., 2012, 2007; Mahieu et al., 2024), which was previously used for the North Atlantic
GEOTRACES (GA03) samples collected at TAG (Buck et al., 2015). This method has been
successfully intercalibrated (Buck et al., 2012, 2016) and allows for an assessment of stronger
iron-binding organic ligands (Rue and Bruland, 1995; Buck et al., 2010; Bundy et al., 2018;
Mahieu et al., 2024). For this approach, samples were aliquoted into carefully conditioned (Mahieu
et al., 2024) flat-bottom Teflon vials (Savillex), buffered with 50 μL of 1.5 M borate buffer (7.5
mM final concentration) to a pH of 8.2 (total scale), and any excess ligands in the samples were
titrated with 15 additions of 0–10 nM $FeCl_3$. The Fe additions were allowed to equilibrate
overnight before adding 25 μM SA to each vial, and SA additions were then equilibrated at least
1 hour before analysis. The measurements were made on a BioAnalytical Systems (BASi)
controlled growth mercury electrode interfaced with an Epsilon E2 analyzer (BASi), using a
deposition time of 120 s. A post-titration spike was used to verify vial conditioning and the
linearity of the final titration points (Mahieu et al., 2024). Titration data were processed using
ECDsoft and ProMCC software (Omanović et al., 2015; Pižeta et al., 2015). We used the complete
complexation fitting method, which fits the data using multiple regression models and allows a
visual verification of results against the titration data.
Reverse titrations were carried out using the competitive ligand 1-nitroso-2-naphthol (NN)
to compete against natural ligands, as has been previously applied to hydrothermal plumes
(Hawkes et al., 2013a). Briefly, 500 μL of 1.5 M trace metal clean borate buffer was added to 150
mL of sample in an acid-cleaned 250 mL LDPE bottle (final concentration of 5 mM) to achieve a
pH of 8.2 (total scale). The buffered sample was pipetted into the preconditioned vials, and 0.5–
40 μM NN was added to compete against natural $L_{Fe}$ complexes. The samples were equilibrated
overnight and the concentration of $Fe(NN)_3$ in each vial was measured by AdCSV after purging
with nitrogen for 300 s using a 797VA Computrace (Metrohm) at the University of South Florida.
For samples with high Fe concentrations (>100 nM) a 5-fold dilution with MQ was conducted to
ensure adequate excess NN (Hawkes et al., 2013a; 2013b; Kleint et al., 2016), and salinity
corrected side reaction coefficients were applied (Gledhill and Van Den Berg, 1994). Previous
work has shown that a 5-fold dilution does not impact the $K_{FeL,Fe'}^{cond}$ but may lead to a slight
overestimation of $L_{Fe}$ (Hawkes et al., 2013b). No dilutions were performed on forward titrations.
The reverse titration data were processed in the software package ECDsoft, and the data were
modeled using a previously published R package (Hawkes et al., 2013a a). Values of [$L_{Fe}$] and
$K_{FeL,Fe'}^{cond}$ were fit to the experimental data where 80% of $i_{pmax}$ was reached using the R package with
values of $\alpha_{Fe'} = 10^{9.8}$ and $\alpha_{FeNN_3} = \beta_{FeNN_3}[NN]^3$ with a $\beta_{FeNN_3}$ value of $5.12 \times 10^{26}$ (Hawkes et al.,
2013a).
### 2.5 Microbial community composition using 16S rRNA gene sequencing
Samples for microbial community composition were taken directly from the carboy for
each incubation. Two liters of water was sampled in a 4 L high-density polyethylene (HPDE)
bottle. Cells were harvested by pressure filtration onto 25 mm diameter sequential 3 and 0.2 μm
pore-size polyethersulfone membrane filters (Supor-200, Pall Corporation) housed in
polypropylene filter holders (Whatman SwinLok) using a peristaltic pump and silicone tubing.
Pump tubing was acid washed with 10% HCl and flushed with ultrapure water between each
sample. The filters were flash frozen in liquid nitrogen in 2 mL gasketed bead beating tubes (Fisher
Scientific). Nucleic acids (DNA) were extracted as using a modified method from Santoro et al.,
242 (2010).
The 16S rRNA gene was amplified in all samples using V4 primers (515F-Y and 806RB,
(Apprill et al., 2015; Parada et al., 2016) following the protocol outlined in Stephens et al., (2020)
and amplicons were sequenced via a 2x250bp MiSeq 500 run at the UC Davis Genome Center.
Resulting sequences were filtered and trimmed with DADA2 (Callahan et al., 2016) and taxonomy
was assigned with the SILVA SSU database (v 138.1, (Quast et al., 2012)) Read counts were
transformed from absolute to relative abundance and taxa were aggregated to the Family level.
The five most abundant families present in each sample were visualized using the 'ggplot2'
package (v. 3.3.5). In order to assess the potential of the observed prokaryotic taxa to produce
siderophores, we downloaded all siderophore biosynthetic gene clusters (BGCs) in the antismash
secondary metabolite database ($n = 7909$) and used text-string matching to compare genera
containing these BGCs to the genera found in our 16S rRNA gene dataset (Blin et al., 2021).
### 3.0 Results
### 3.1 Experimental context
The incubations presented here captured distinct physical (buoyant/neutrally buoyant) and
chemical (reducing/oxidized) stages of the hydrothermal plumes (Gartman and Findlay, 2020)
across a range of systems with high and low Fe vent fluids (Table 1). We used the concentration
of Mn as a quasi-conservative tracer (Field and Sherrell, 2000) of mixing between vent fluid
endmembers (Table 1) and background deep ocean concentrations in the region (~0.1 nM; Hatta

et al., (2015)). The concentration of Mn was stable over the days to weeks of incubation time in all experiments, with all Mn observed in the soluble fraction (Fig. S2-S5). In field measurements from GA13, Mn was shown to behave similarly to helium-3 ($^3$He), indicating that over the spatial and temporal scope of these experiments it is a valid proxy for dilution (Lough et al., 2023). We used a dilution factor of 10,000 to define the threshold between the buoyant plume and the neutrally buoyant plume (Lupton et al., 1985), but also report the potential density anomaly ($\sigma_0$) from each cast (Figure S1). The concentrations of TDFe were considered representative of the sum of dissolved and particulate Fe in each incubation, following previous work in near-field hydrothermal systems (Revels et al., 2015; Lough et al., 2017). We present the incubation results in order from the closest to the vent source (high Fe, reducing, near-field) to the furthest away from the vent source (lower Fe, oxidized, far-field).

| | Lucky Strike[a] | Rainbow (near) | Rainbow (far) | TAG | Deep Atlantic |
|---|---|---|---|---|---|
| Max T (°C) | 260 | 365 | 365 | 363 | 2.5-4.5 |
| pH | 4.3 | 2.8 | 2.8 | 3.1 | 7.8 |
| $H_2S$ (mM) | 3 | 1 | 1 | 3.5 | 0 |
| Mn (µM) | 150 | 2,250 | 2,250 | 710 | $\sim10^{-4}$ |
| Fe (µM) | 240 | 24,000 | 24,000 | 5,170 | $\sim10^{-3}$ |
| $H_2S$:Fe | 12.5 | 0.04 | 0.04 | 4.1 | 0 |
| Initial Mn (nM)[b] | 10 | 500 | 2.5 | 35 | N/A |
| Dilution factor | 15,000 | 4,500 | $9\times10^5$ | $\sim21,000$ | N/A |
| Plume stage[c] | NBP | BP | NBP | NBP | N/A |

Table 1. Data from Douville et al., (2002) of vent fluid end member chemical characteristics for the three vent systems studied in these incubation experiments. Presented are the dissolved Mn, Fe, $H_2S$ concentrations, the maximum temperature, and pH of the vent fluid endmembers. [a]Lucky Strike end member data was compiled from Pester et al., (2012) using averages of the vent sites US4 and IsabelMeSH, determined to be the closest systems to the location in which the incubation was initiated. [b]The concentration of Mn observed in each incubation, used to calculation the dilution factor for each experiment. [c]The plume stage is indicated as buoyant plume (BP) or neutrally buoyant plume (NBP) using a threshold dilution factor of 10,000:1 based on Lupton et al. (1985).

*3.2 Rainbow near-field incubation – a high Fe, reduced, buoyant plume*

The Rainbow vent system has the highest vent fluid Fe concentrations of any documented system in the global ocean (German et al., 2025). This incubation was conducted directly at the vent site (near-field), and incubated both unfiltered and filtered (<0.2 µm) plume water for 7 days. The dilution factor estimated from published endmember data was ~ 4,500 (Table 1; (Douville et al., 2002). A positive temperature (~0.2 °C) anomaly, negative density anomaly (~0.02 kg m$^{-3}$), and ORP anomaly observed at the depth of sampling confirmed that this incubation was initiated from water within a reducing buoyant plume (Fig. S1c, d). The unfiltered incubation at Rainbow had high TDFe concentrations of 5094 ± 93 nM (1SD, *n*=5; Fig. 2a) and initial dFe concentrations of 361.3 ± 0.5 nM (Fig. 2c); Fe(II) concentrations ranged from 40 to ~70 nM within the plume (González-Santana et al., 2023). The filtered incubation had TDFe concentrations of 356 ± 9 nM (1SD, *n*=5; Fig 2b) with an initial dFe concentration of 311.5 ± 6.1 nM (Fig. 2d). High sFe concentrations of 70.9 ± 4.6 and 94.1 ± 5.9 nM were also observed at the onset of the unfiltered and filtered incubation, respectively, but the dissolved phase was still dominated by colloids in the unfiltered (80%) and filtered (70%) incubations (Table S2). The $\delta^{56}$dFe of the initial plume samples were distinct between the unfiltered and filtered incubations, with values of -7.35‰ and 0.76‰, respectively (Fig. 2c, 2d; Table S6).

Rapid changes in the physicochemical partitioning of Fe were observed in the first 24 hours of the incubation. Broadly, dFe concentrations declined by 33% and 70% in the unfiltered and filtered treatments over the 6-days of incubation, respectively (Fig. 2c, 2d; Table S2), but this decline was punctuated with deviations from this trend in the early hours of incubation. In the unfiltered incubation, sFe concentrations more than doubled in the first 12 h, increasing from 70.9 ± 4.6 to concentrations as high as 196.8 ± 10.1 nM (Fig. 2c; Table S2), coinciding with large increases in $H_2S$ (Fig. S7; Text S1). By 24 h, sFe concentrations had declined 85% from its peak to 29.7 ± 3.8 nM concomitant with an increase in dFe to 415.8 ± 3.1 nM (Fig. 2c). In the filtered incubation, sFe concentrations showed high variability over the first 24 h, before declining to 1.91 ± 0.12 by the end of the incubation (Fig. 2d). After the first 24 hours of the incubation the colloidal fraction of the dFe pool in both the filtered and unfiltered treatments stabilized at ~90-97% (Fig. 2c, 2d; Table S2). The isotopic composition of the dFe pool also changed over the course of the incubation. During the first 12 h, the $\delta^{56}$dFe in the unfiltered incubation decreased from 0.73‰ to -0.16‰ along with the large changes observed in the particulate and dFe phases. The final measurement taken on day 5 showed an enrichment of $\delta^{56}$dFe to 1.17‰ (Figure 2c; Table S6). The $\delta^{56}$dFe in the filtered incubation remained relatively stable throughout, with an average value of -7.31 ± 0.31‰ (2SD, $n$=3).

Dissolved and soluble Fe-binding organic ligand concentrations were elevated at the onset of the Rainbow incubation experiment, with different dynamics in filtered and unfiltered treatments. The $dL_{Fe}$ measured initially were 64.4 ± 1.4 nM and 45.8 ± 0.8 nM in the unfiltered and filtered treatments, respectively, with conditional stability constants ($K^{cond}_{FeL,Fe'}$) ranging from 11 to 12.5. The initial concentration of $sL_{Fe}$ were 5.37 ± 0.53 and 3.57 ± 0.40 in the unfiltered and filtered treatments, respectively, with weaker $K^{cond}_{FeL,Fe'}$ than those observed for $dL_{Fe}$ ($K^{cond}_{FeL,Fe}$=10.0–10.8; $t$-test, $p$=0.06 and 0.02, respectively; Fig. 2e, 2f; Table S2). The concentration of $dL_{Fe}$ declined 42% and 25% over 7 days in the unfiltered and filtered incubations, respectively (Fig. 2e, 2f). In contrast, concentrations of $sL_{Fe}$ in the unfiltered treatment increased by nearly 3-fold during the first 24 h of the incubation concomitant with the increases of sFe. The $sL_{Fe}$ subsequently declined close to initial concentrations of 5.23 ± 0.16 nM by day 2, where they remained relatively stable until the final sample taken on day 7 where a 2-fold increase to 8.57 ± 0.49 nM was observed independent of sFe (Fig. 2c, 2e). In the filtered treatment, $sL_{Fe}$ similarly increased nearly 3-fold from their initial concentrations of 3.57 ± 0.49 nM to 9.27 ± 0.13 nM by day 2, decreasing to 5.64 ± 0.35 nM for the remainder of the incubation (Fig. 2d, 2f). The conditional stability constants $sL_{Fe}$ sampled in the first 24 h of both treatments were typically weaker ($K^{cond}_{FeL,Fe}$=10.0–10.8; Fig. 2e, 2f; Table S2) than the ligands measured during the remainder of the incubation ($K^{cond}_{FeL,Fe'}$=11.0–12.2).

The microbial community composition was sampled in the initial (day 0) and final (day 7) timepoints of the incubation for the unfiltered treatment for two separate size fractions (0.2 and 3 μm). Overall, both size classes displayed a similar community composition and response throughout the incubation. Both size-fractions on day 0 and day 7 samples were dominated by the sulfur-oxidizing bacteria *Sulfurimonas* (family *Sulfurimonadaceae*). This family comprised ~75% of the microbial community in both size fractions (Fig. S8). The SUP05 cluster (*Gammaproteobacteria* belonging to the *Thigolobaceae* family; Fig S8) are also known sulfur-oxidizers that were abundant to a lesser degree (Shah et al., 2017). The ammonia-oxidizing archaea *Nitrosopumilaceae* (Könneke et al., 2005) were also observed at the start of the incubation but decreased in relative abundance by the end of the experiment (Fig. S8).

# Rainbow near-field

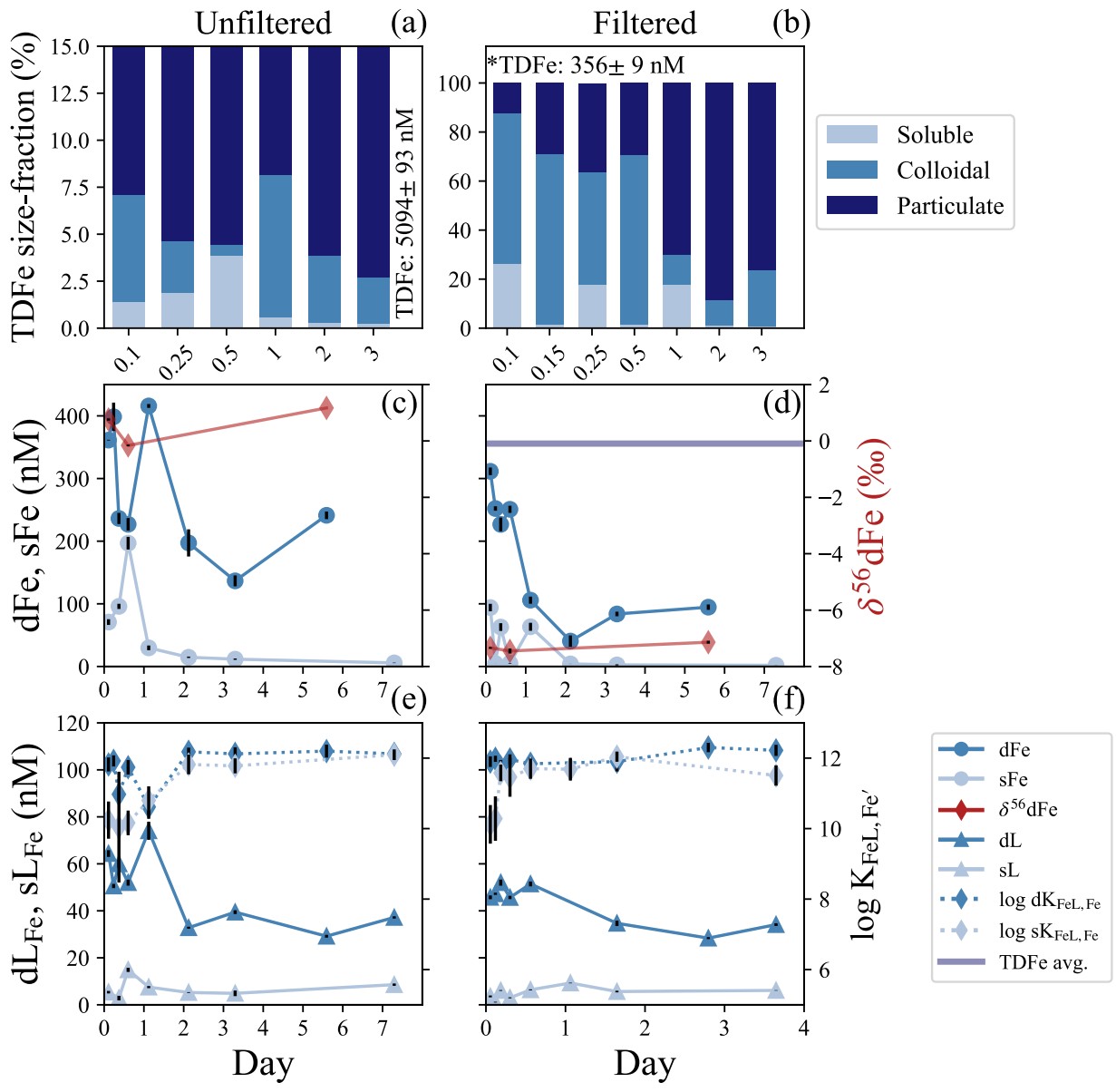

Figure 2. Rainbow near-field incubation overview for unfiltered (left column) and filtered (<0.2 μm, right column) treatments. (a, b) Particle phase composition for each timepoint displaying dissolved ('d'), soluble ('s'), and total dissolvable ('TD'), reporting the average TDFe of the incubation (1SD, $n$=5, *$n$=5, one sample removed; Table S2). (c, d) sFe (light blue circles) and dFe (dark blue circles), and the average TDFe concentrations of the incubation (blue line); the TDFe concentration for panel (c) is off the scale. The $\delta^{56}$dFe composition for selected samples (red diamonds) is plotted on the secondary axis. (e, f) Concentration of dL$_{Fe}$ (dark blue triangles) and sL$_{Fe}$ (light blue triangles) and the respective log $K_{FeL,Fe'}^{cond}$ (diamonds, dashed lines) are plotted on the secondary axis.

## 3.3 TAG incubation – a high Fe, reduced, neutrally buoyant plume

The TAG vent field sits ~1,300 m deeper than Rainbow at 3,620 m water depth (Tivey, 2007), and is one of the most studied vent systems along the MAR. This was a 6-day incubation that contained filtered and unfiltered treatments. The dMn concentrations at the onset of the incubation were 38.5 ± 1.4 nM and 31.8 ± 0.3 nM in the unfiltered and filtered incubations,

respectively, resulting in an estimated dilution factor between ~19,000–22,500 from the published endmember data (Table 1). Across the ~100 m depth range of the plume, there was an LSS and ORP anomaly and a small (~0.1 °C) negative temperature anomaly consistent with a reducing neutrally buoyant plume (Fig S1e, S1f). The initial dFe concentrations at the start of the incubation were 188 ± 2.3 nM and 82.4 ± 0.6 nM dFe in the unfiltered and filtered treatments, respectively (Fig. 3b, 3c; Table S3). The Fe(II) concentrations from within the plume ranged from 4–10 nM (González-Santana et al., 2023). The average TDFe at TAG (unfiltered treatment) was 211 ± 9 nM (1SD, $n$=2), suggesting ~80% of the vent end-member was present at the start of the incubation, based on the dilution factor calculated from dMn. Approximately 90% of the total TDFe was within the dissolved phase (Fig. 3a, 3b). Soluble Fe concentrations were 61.5 ± 8.2 nM and 19.1 ± 2.3 nM in the unfiltered and filtered incubations, resulting in a colloidal fraction of 67% and 77%, respectively. The $\delta^{56}$dFe at the start of the incubation in the unfiltered and filtered treatments were 0.58‰ and -1.43‰, respectively. The low $\delta^{56}$dFe value as observed in the filtered treatment is consistent with previously reported $\delta^{56}$dFe value in the water column from the TAG hydrothermal plume (-1.35‰; Conway and John, (2014).

The high concentrations of dFe declined over the 6 days of the incubation and the particulate fraction increased (Fig 3a, 3b). By the end of the incubation, dFe concentrations in both unfiltered and filtered treatments had declined by 67% and 83%, respectively (Fig. 3c, 3d). The high sFe concentrations observed at the start of the incubation declined in parallel to the dFe, decreasing by 77% and 67% in the unfiltered and filtered treatments, respectively. The observations of $\delta^{56}$dFe in the unfiltered and filtered treatments for this incubation showed similar patterns to those in the Rainbow near-field experiment. In the unfiltered treatment, the $\delta^{56}$dFe decreased to 0.09‰ in the first 24 h, but in the final sample taken on day 6 showed a significant enrichment to 3.57‰ (Fig. 3c; Table S6). Similarly, the $\delta^{56}$dFe values of the filtered treatment remained constant (-1.37 ± 0.10‰, 2SD, $n$=3) over the 6-day incubation.

The Fe-binding ligands observed at TAG displayed similar characteristics to those in the Rainbow near-field incubation. The $dL_{Fe}$ concentrations at the onset of the incubation were 18.4 ± 1.1 nM and 5.97 ± 0.25 nM in the unfiltered and filtered treatments, respectively (Fig. 3e, 3f). Over the 6-day incubation, $dL_{Fe}$ concentrations declined by 64% in the unfiltered treatment. In contrast, the $dL_{Fe}$ concentrations in the filtered treatment increased to a maximum of 11.80 ± 0.40 nM in the first 24 h and then declined to a final concentration of 7.03 ± 0.39 nM, resulting in an overall increase of ~15% over the 6 days. In the unfiltered treatment, $sL_{Fe}$ concentrations decreased from 8.27 ± 0.76 nM to 4.48 ± 0.27 nM in the first 24 h, remained constant until day 3 and then increased to a final concentration of 7.68 ± 0.09 nM, close to initial concentrations (Fig. 3e). In the filtered treatment on the other hand, $sL_{Fe}$ concentrations declined by 63% during the 6-day incubation. The conditional stability constants of both $dL_{Fe}$ and $sL_{Fe}$ pools were relatively weak throughout the incubation, with average $\log K^{cond}_{FeL,Fe'}$ = 10.76 ± 0.70 (1SD, $n$=7) and 10.87 ± 0.79 (1SD, $n$=9) in the unfiltered and filtered treatments, respectively. Stronger ligands were observed on average in the $sL_{Fe}$ pool of the unfiltered (11.20 ± 0.74 (1SD, $n$=8) and filtered treatments (11.51 ± 0.53 (1SD, $n$=7), respectively (Figure 3e, 3f; Table S3). Although on average the $sL_{Fe}$ had larger stability constants relative to those in the dissolved phase, no significant differences were observed between the two size fractions in the unfiltered (*t-test, p*=0.28) and filtered (*t-test, p*=0.08) treatments.

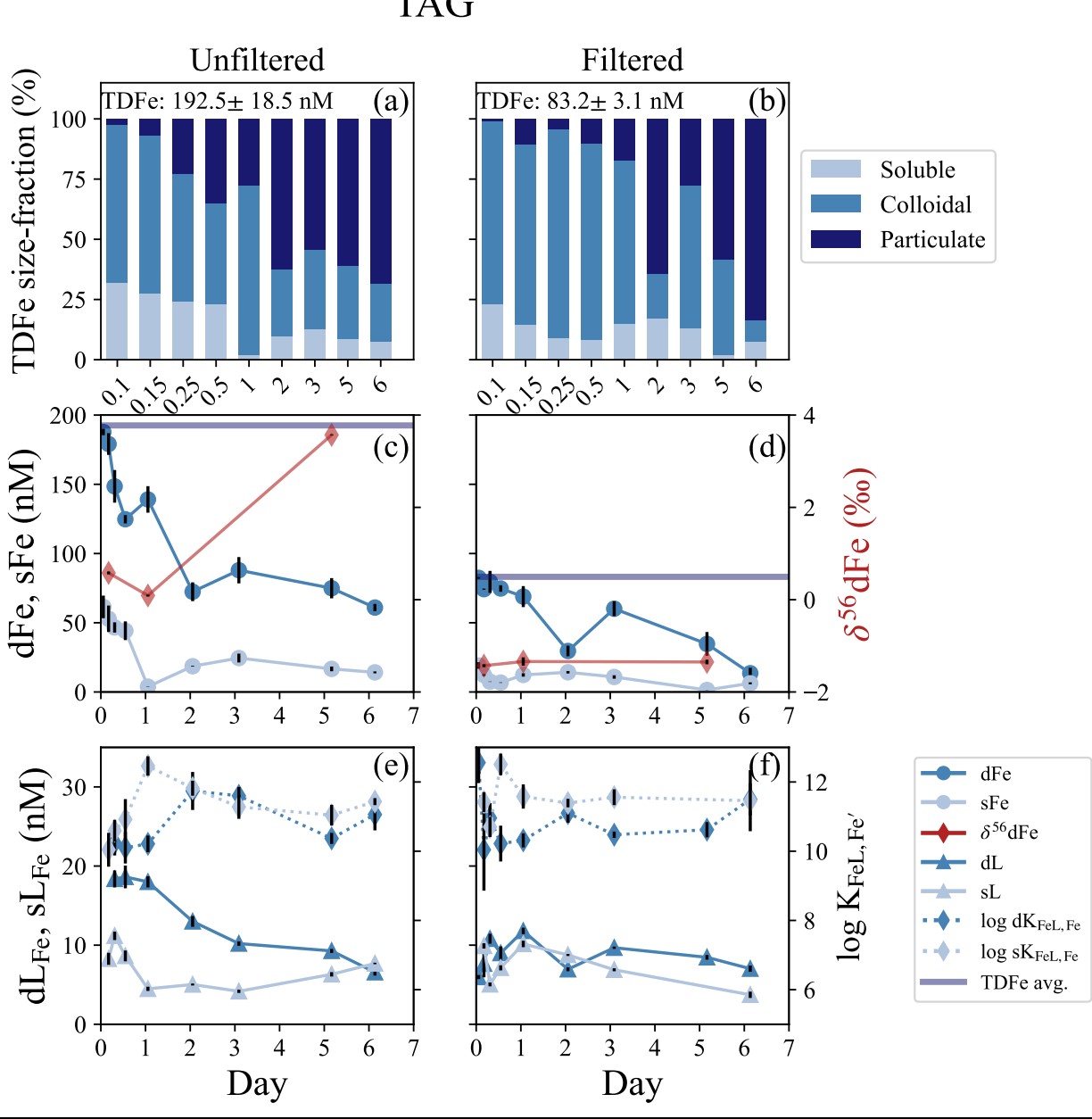

Figure 3. TAG incubation overview for unfiltered (left column) and filtered (<0.2 μm, right column) treatments. (a, b) Particle phase composition for each timepoint displaying dissolved ('d'), soluble ('s'), and total dissolvable ('TD'), reporting the average TDFe of the incubation (1SD, $n$=2). (c, d) sFe (light blue circles) and dFe (dark blue circles), and the average TDFe concentration (blue line) of the incubation. $\delta^{56}$dFe composition for selected samples (red diamonds) is plotted on the secondary axis. (e, f) Concentration of $dL_{Fe}$ (dark blue triangles) and $sL_{Fe}$ (light blue triangles) and the respective log $K_{FeL,Fe'}^{cond}$ for each measurement (diamonds, dashed lines) is plotted on the secondary axis.

*3.4 Lucky Strike incubation – a low Fe, reduced, neutrally buoyant plume*

The Lucky Strike vent field is located atop a seamount over the Azores hotspot at relatively shallow depths (1,600-1,700 m; Pester et al., 2012). Only an unfiltered treatment was incubated from Lucky Strike and the experiment lasted 22 days. An estimated dilution factor of ~15,000 was calculated from dMn endmember concentrations (Table 1) and the bottles were closed within a small LSS and ORP anomaly with stable density, indicating the incubation was started in a reducing neutrally buoyant plume (Fig S1a, S1b). The average TDFe concentration of the Lucky Strike incubation was $20.1 \pm 1.8$ nM (1SD, *n*=4), with initial dFe concentrations of $15.1 \pm 1.17$ nM and sFe of $4.42 \pm 0.05$ nM, resulting in ~80% of the Fe within the dissolved phase and 71% of the dFe in colloidal form (Fig. 4b; Table S4). The Fe(II) concentrations from this plume ranged from 0.4–0.8 nM (González-Santana et al., 2023). The Fe-binding ligand concentrations at the start of the incubation were relatively elevated, with initial $dL_{Fe}$ concentrations of $5.40 \pm 0.12$ nM. These ligands appeared to be entirely within the soluble phase, with $sL_{Fe}$ concentrations equal to $6.00 \pm 0.60$ nM at the onset of the incubation (Fig. 4c; Table S4).

Lucky Strike was the longest incubation experiment, and enabled observations of both early and long-term particle formation dynamics. Initially, dFe concentrations declined ~30% over the first week and particulate Fe concentrations increased, with the lowest concentration of dFe ($10.4 \pm 0.81$ nM) observed on day 7 (Fig. 5b). The soluble and dissolved Fe-binding ligands also declined in the first 48 h of the incubation but then increased from days 3 to 7, with the $sL_{Fe}$ representing 100% of the dissolved ligand pool by day 7 (Fig. 4c). In subsequent weeks, $dL_{Fe}$ and dFe continued to increase, while particulate Fe decreased. The final dFe concentration observed at 22 days was $18.3 \pm 1.4$ nM, roughly 20% higher than the initial dFe concentrations, and sFe increased to concentrations observed at the start of the incubation of $4.55 \pm 0.32$ nM. By the end of the incubation nearly all the TDFe was in the dissolved fraction and dominated by colloids (Fig. 4a). Across this incubation the Fe-binding ligands in the soluble fraction were significantly stronger than those observed in the dissolved fraction (*t-test, p=0.005, n=7*).

The Lucky Strike incubation showed a more dramatic shift in the microbial community composition compared to the Rainbow near-field incubation, and a distinct difference in the community composition between the two size-fractions. The larger 3 µm size fraction was dominated by sulfur-oxidizing bacteria *Sulfurimonas* (*Sulfurimonadaceae*) and SUP-05 (*Thioglobaceae*), as in the Rainbow near-field incubation (Fig. S9). However, the smaller size-fraction (0.2 µm), contained both sulfur-oxidizers, and the ammonia-oxidizer *Nitrosopumilaceae*. After 14 days, the 3 µm community became dominated by bacterial families *Alcanivoraceae*, *Oleiphilaceae*, and *Sphingomonadaceae*, typically associated with hydrocarbon degradation (Fig. S9). No data were available for the smaller size fraction at this timepoint.

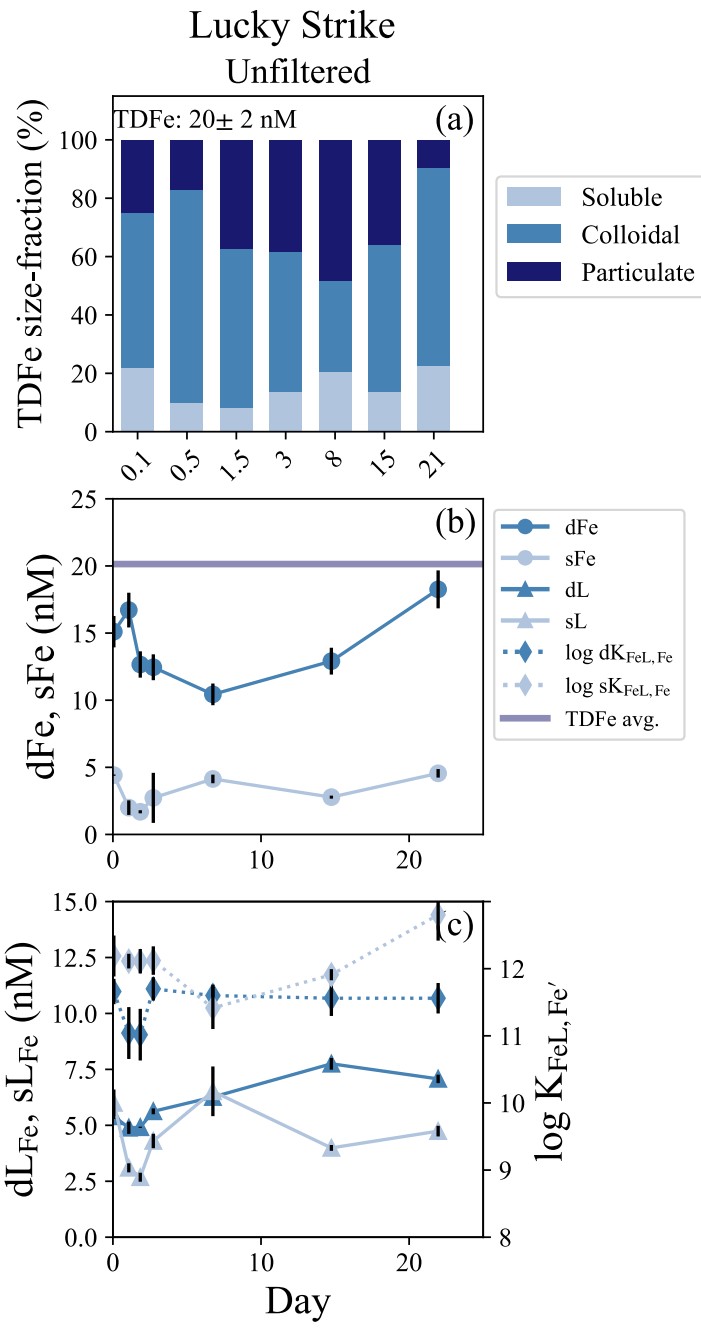

Figure 4. Lucky Strike incubation overview. (a) Particle phase composition of each timepoint in the incubation with dissolved ('d'), soluble ('s'), and total dissolvable ('TD') data, reporting the average TDFe concentration of the incubation (1SD, $n$=4). (b) sFe (light blue circles) and dFe (dark blue circles) concentrations and the average TDFe (blue line). (c) Concentration of $dL_{Fe}$ (dark blue triangles) and $sL_{Fe}$ (light blue triangles) and the respective $\log K_{FeL,Fe'}^{cond}$ displayed on the secondary axis (diamonds, dashed lines).

### 3.5 Rainbow far-field incubation – an oxidized neutrally buoyant plume

A second incubation was conducted near Rainbow but far-field of the vent, ~10 km southwest of the location of the near-field incubation. This experiment consisted of only an unfiltered treatment and was incubated for 19 days. The dMn concentration at the incubation onset

was $2.5 \pm 0.1$ nM, resulting in a calculated dilution factor of $\sim9.0 \times 10^5$ based on vent fluid end-member data (Table 1). There was a small LSS signal but no ORP anomaly indicating the incubation was started within an oxidized neutrally buoyant plume (Fig. S1g, S1h). The initial dFe and sFe concentrations were $3.42 \pm 0.26$ nM and $0.88 \pm 0.04$ sFe, respectively, and 74% of the dFe was colloidal (Table S5). The average TDFe concentration of the incubation was $25.5 \pm 5.6$ nM (1 SD, $n$=3), indicating $\sim$10% of the Fe was in the dissolved phase at the start of the incubation (Fig. 7a). Fe-binding ligand concentrations at the start of the incubation were also low relative to the other experiments, with $1.55 \pm 0.12$ nM in the dissolved phase and soluble ligands were below the limit of detection (Table S5).

Twenty-four hours into the experiment, soluble ligand concentrations began to increase and continued to do so over the next two days, appearing to modify the Fe particulate pool (Fig. 7c). The dFe increased from $2.96 \pm 0.23$ nM to $8.52 \pm 0.65$ nM between day 1 and day 7, and dL$_{Fe}$ increased from $2.90 \pm 0.11$ nM to $5.41 \pm 0.17$ nM. Little change was observed in sFe concentrations over this period (Fig. 5b; Table S5). Over the subsequent week, the sFe concentrations increased to $7.95 \pm 0.53$ nM by day 14, encompassing the entirety of the dFe pool (Fig. 5a). For the remainder of the incubation the colloidal fraction remained low, comprising 20% of the dFe pool in the final sample at day 19 (Fig. 5b; Table S5). Throughout this incubation there was no significant difference observed in the binding strength of ligands between the dissolved and soluble pools (*t-test, p=0.21, n=7*).

The microbial community composition was sampled initially and again on day 15. The community shifted substantially over this period, similar to the observations at Lucky Strike. At the onset of the incubation, the 3 μm size-fraction was again dominated by sulfur-oxidizers *Sulfurimonadaceae* and *Thioglobaceae*, (Fig. S10). The smaller 0.2 μm size-fraction contained sulfur-oxidizers, but also substantial fractions of *Nitrosopumilaceae* and nitrite-oxidizers *Nitrospinaceae*. Two weeks into the incubation, following the observed ligand production and particulate Fe mobilization, the microbial community in both 3 and 0.2 μm size-fractions shifted toward dominance by the hydrocarbon degraders *Oleiphilus, Sphingorhabdus,* and *Alcanivorax* (Fig. S8 Fig. S10).

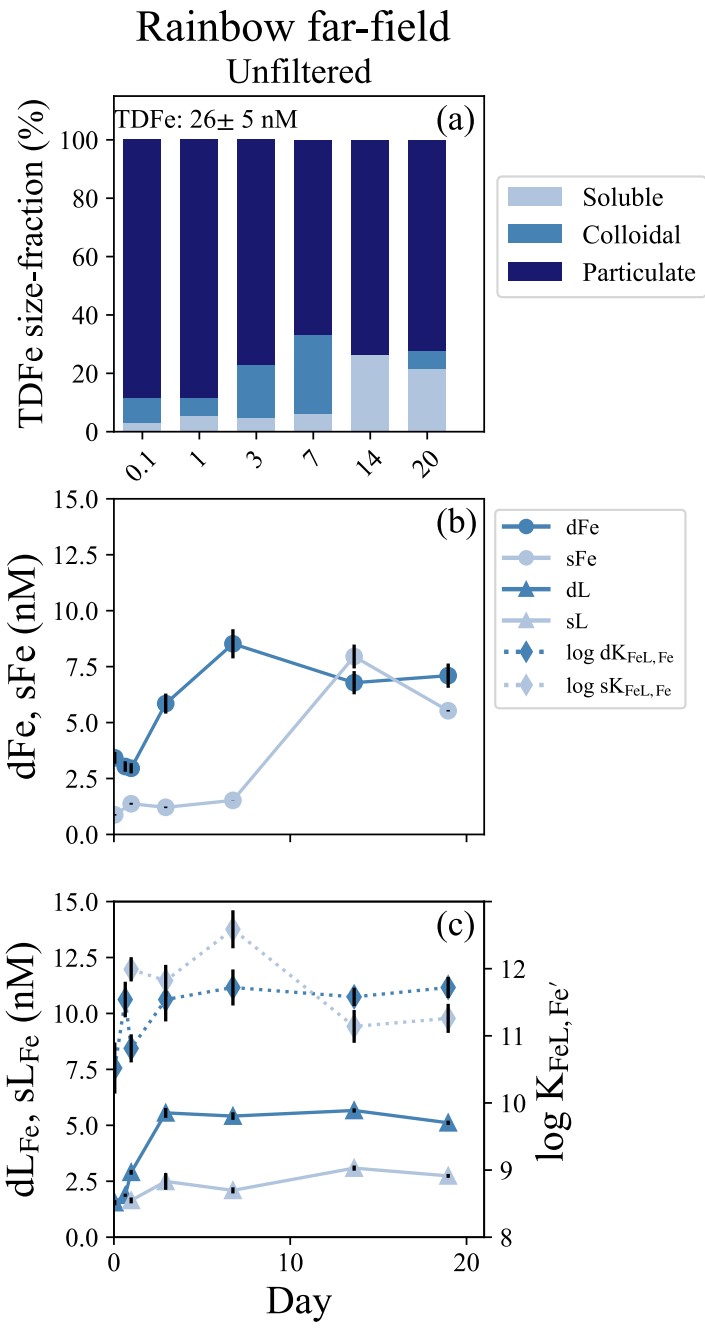

Figure 5. Rainbow far-field incubation overview. (a) particle phase composition of each timepoint in the incubation with dissolved ('d'), soluble ('s'), and total dissolvable ('TD') data, and reporting the average TDFe concentration of the incubation (1SD, $n$=4). (b) sFe (light blue circles) and dFe (dark blue circles) concentrations. (c) concentration (left axis) of $dL_{Fe}$ (dark blue triangles) and $sL_{Fe}$ (light blue triangles) and the respective log $K_{FeL,Fe'}^{cond}$ for each measurement (right axis; diamonds, dashed lines).

### *4.0 Discussion*

### *4.1 Sources and sinks of Fe-binding ligands within early plume systems*

The three near-field incubations of this study show that hydrothermal vents along the Mid Atlantic Ridge are a local source of $L_{Fe}$. Dissolved Fe-binding ligand concentrations at the onset

of each near-field incubation were observed to be well above the background concentrations of
~2–3 nM observed throughout the water column of the deep Atlantic (Buck et al., 2015; Gerringa
et al., 2015). Vent systems are known to host vibrant communities of chemoautotrophs (Sander
and Koschinsky, 2011; Sylvan et al., 2012) and the diffusive flow regions surrounding the vents
have been identified as a potential source of DOC and Fe-binding ligands in other studies (Lang
et al., 2006; Bennett et al., 2008; Hawkes et al., 2013b; Kleint et al., 2016). The high concentration
and low conditional stability constants (log $K_{FeL,Fe}^{cond}$ = 10–11) observed in the early samples of many
of the near-field incubations suggest that this ligand pool may be dominated by non-specific Fe-
binding ligands such as exopolysaccharides (Hassler et al., 2011; Zykwinska et al., 2019) or other
organic acids (Konn et al., 2009). The ligands could be a product of the serpentinization reactions
within the vent system (organic acids; soluble fraction) or the biological communities surrounding
them (EPS; colloidal fraction) that are entrained in the forming plume (Bennett et al., 2008;
Hawkes et al., 2013b; Sander and Koschinsky, 2011).
509        Although these experiments show these vents are a large source of ligands, our
observations show early particle formation in the vent plumes may also be a major ligand sink. In
both near-field incubations, at TAG and Rainbow, initial dFe concentrations were much higher
than $dL_{Fe}$, and yet $dL_{Fe}$ declined along with dFe in a nearly 1:1 ratio as the experiments became
progressively more particle enriched (Fig. 2c-f, 3c-f). This suggests that as Fe particles form over
time, the dissolved ligand pool is scavenged or aggregated along with dFe. The constant $\delta^{56}$dFe in
these week-long incubations, despite 71-77% declines in dFe, is inconsistent with isotopic
fractionation expected by Fe(II) oxidation and precipitation alone (Fig. 2d, 3d; Section 4.2). Thus,
the loss of dFe and formation of particulate Fe over time may instead represent the coagulation of
colloidal ligand-bound Fe and/or Fe-(oxyhydr)oxides, analogous to observed flocculation of Fe
with organic matter during estuarine mixing (Boyle et al., 1977; Buck et al., 2007; Bundy et al.,
2015), which does not substantially fractionate dFe isotopes (Escoube et al., 2009; Zhang et al.,
521   2015).
522        Another consistent observation from these experiments was the production of soluble Fe-
binding ligands in the unfiltered treatments of each incubation. Soluble Fe-binding ligands were
more resistant to aggregation and losses than ligands in the colloidal fraction. In the two near-field
and high-Fe reducing incubations conducted at Rainbow and TAG, the unfiltered treatments
showed a 42% and 64% decrease in $dL_{Fe}$ between the first and final measurements. In comparison
there was a 60% increase in $sL_{Fe}$ in the near-field Rainbow experiment and a 15% decrease in the
TAG experiment (Figure 2e, 4e; Table S2, S3). Whereas aggregation appeared to dominate $dL_{Fe}$
in the early days of the reducing plume incubations of Rainbow, TAG, and Lucky Strike, $sL_{Fe}$
concentrations increased within days of experiment initiation in all cases. Evidence of siderophore
production has been inferred in hydrothermal plumes in the Guaymas Basin through the
upregulation of siderophore biosynthesis and transporter genes (Li et al., 2014) and siderophores
have been identified in field samples from the same plumes along the GA13 cruise (Hoffman et
al., 2024). Siderophores should fall within the soluble fraction of dFe when bound to Fe (Vraspir
and Butler, 2009), and are known to solubilize Fe from larger size fractions (Borer et al., 2005;
Fishwick et al., 2014). Production of $sL_{Fe}$ coincided with a large shift in the microbial community
from one dominated by sulfur-oxidizers (*Sulfurimonadaceae* and *Thioglobaceae*), ammonia-
oxidizers (*Nitrosopumilaceae*), and nitrite oxidizers (*Nitrospinaceae*, Fig. S8, S9, S10) to a
community dominated by families associated with hydrocarbon degradation (*Alcanivoraceae*,
*Oleiphilaceae*, and *Sphingomonadaceae* (Yakimov et al., 1998; Golyshin et al., 2002; Kertesz et
al., 2019)). Most of the families that increased their share of total 16S rRNA gene amplicons by

the end of these two incubations with microbial data had putative microbial community members capable of producing siderophores, although the 16S data alone cannot resolve whether siderophore producing members were present. A heavier $\delta^{56}$dFe isotopic signature in the final samples of the unfiltered treatments at Rainbow near-field and TAG is also consistent with the possibility of siderophore production, as the complexation of dFe by siderophores would be expected to impart a heavier fractionation signature on the dFe pool (Dideriksen et al., 2008).

Together, the observations from these experiments and from a complimentary water column study (Hoffman et al., 2024) provide compelling evidence for microbial ligand production as a mechanism for the stabilization of Fe in these hydrothermal plumes. Experiments that were initiated closer to the vent source initially had higher concentrations of weaker ligands, predominantly in the colloidal fraction, that aggregated into the particulate phase as dFe declined in the first hours of the experiment. In the experiments further from the vent source and over the timescale of days to weeks, the ligand pool became stronger overall and stronger ligands were produced over time in the soluble fraction in the unfiltered treatments. In general, our experiments demonstrated that there is a transition in the Fe-binding ligand pool in both time and space with distance from the vent source. We postulate the temporal transition from a hydrothermal plume that contains reduced substrates and Fe(II) to one that is oxidized and dominated by Fe(III) may result in a microbial community response that promotes ligand (e.g., siderophore) production and increases the residence time of dFe in the plume.

### 4.2 Particle-dissolved exchange in hydrothermal plumes

Hydrothermal plumes are regions of extreme chemical disequilibria, and the mixing of cold deep ocean waters and hot metal-rich hydrothermal fluid results in huge changes in Fe solubility that results in rapid particle formation. Our near-field experiments conducted at Rainbow, TAG, and Lucky Strike, all had initially high-Fe and reducing conditions that resulted in the rapid formation of colloids and particles. There was evidence of physicochemical exchange between newly formed particles and the dissolved phase within hours in the high-Fe reducing plumes of TAG and Rainbow, and over days to weeks in the Rainbow far-field and Lucky Strike incubations, potentially mediated by ligand production (see section 4.1). The physicochemical partitioning of Fe in combination with dFe isotope provides additional context into the possible mechanisms responsible for the temporal transformations of Fe within early plumes.

Initial conditions in both TAG and Rainbow near-field plumes showed large differences in dFe fractionation between the filtered and unfiltered treatments, with the filtered treatments having significantly lighter (−1.43‰, TAG; −7.34‰, Rainbow) values than published vent fluid endmembers of TAG and Rainbow (−0.15‰ and −0.14‰, respectively; (Severmann et al., 2004)). These depleted $\delta^{56}$dFe values in both experiments are most likely a result of the partial oxidation of Fe(II) and precipitation of Fe-(oxyhydr)oxide particles at the start of the incubations that was interrupted by the filtering process (Lough et al., 2017; Klar et al., 2017; González-Santana et al., 2021; Wang et al., 2021). The light isotopic values observed in the Rainbow near-field experiment likely reflect its proximity to the vent source, the very high Fe/H$_2$S ratio of the vent fluids (Table 1, Severmann et al., 2004), and high Fe(II) oxidation rates (0.29 h; Field and Sherrell, 2000; González-Santana et al., 2021) relative to other vent sites (e.g., Klar et al., 2017; Wang et al., 2021). Laboratory experiments show the preferential incorporation of heavy Fe isotopes into Fe-(oxyhydr)oxides, which leaves the remaining dFe isotopically light (Welch et al., 2003). For example, analyses of hydrothermal particles by Revels et al. (2015) and Severmann et al. (2004) showed $\delta^{56}$Fe enrichment of particles of ~0.20‰ in the TAG plume and 0.24-1.29‰ (average

~0.90‰) in the Rainbow buoyant plume. These observations were attributed to the partial oxidation of Fe(II) and precipitation of the Fe(III) that formed as the plume was physically transported. Initial $\delta^{56}$dFe values in the unfiltered treatment (collected within ~3 h) showed a similar enrichment of +0.58‰ for TAG and +0.76‰ for Rainbow, significantly heavier than in the companion filtered treatments and likely reflect the contribution of suspended colloids to the dFe pool.

The evolution of $\delta^{56}$dFe over the week-long incubations at both TAG and Rainbow near-field were also suggestive of the influence of organic ligands on the dFe pool. In the filtered treatments for both experiments, the initial conditions suggested Fe(II) was present and was partially oxidized, but the continued oxidation of Fe(II) was not supported by the isotopic evidence and instead a fairly static isotopic ratio was maintained despite large declines in dFe concentrations (Fig. 2d, 3d). These observations are distinct from the only other temporal studies on hydrothermal plume water, which observed a negative $\delta^{56}$dFe signal (-5‰) over a 24 h period from subsamples of a Niskin bottle (Lough et al., 2017) that was interpreted as continued oxidation of Fe(II) over that time period. Our observations from the filtered treatments, however, indicated dFe was in a form resistant to further fractionation, such as being complexed to organic ligands (see section 4.1). The first 12–24 h of the unfiltered treatment did show some initial negative fractionation (TAG: -0.49‰; Rainbow: -0.92‰; Table S6), which could be suggestive of oxidative fractionation at these short timepoints, but this trend reversed dramatically towards the end of the incubations.

The final timepoint of both unfiltered treatments showed substantial enrichment (up to +3.57‰ at TAG and +1.17‰ at Rainbow) of the $\delta^{56}$dFe by the end of the incubation (Fig 2a, 3a). Simple (abiotic) dissolution of Fe-(oxyhydr)oxide particles has a relatively small isotope effect (< 0.1‰; (Skulan et al., 2002) and ligand-controlled dissolution of Fe-(oxyhydr)oxides is thought to enrich dFe isotopes by up to ~0.5–0.6‰ in solution (Wiederhold et al., 2006; Dideriksen et al., 2008). Although continuous inorganic scavenging could also contribute to $\delta^{56}$dFe enrichment by up to ~0.3‰ (John and Adkins, 2012), soluble ligand production observed over this same period (Figure 3e, 3f; Section 4.2) supports ligand-mediated dissolution. Similar enrichments of $\delta^{56}$dFe have been observed in surface waters of the Southern Ocean (Ellwood et al., 2020; Sieber et al., 2021). These heavy isotopic signatures have been interpreted as resulting from rapid recycling of Fe between the dissolved and cellular particulate pools beyond what might be modelled by simple uptake or organic complexation alone. Siderophores or other strong ligands can provide a mechanism for rapid recycling of Fe between dissolved and cellular Fe pools (Tortell et al., 1999). Although there is very little experimental data on the fractionation of Fe by microbial communities, the heavier fractionation observed in the unfiltered treatments support the organic complexation of the dFe pool, which may have been amplified by uptake of Fe by the bacteria community and/or rapid recycling of dFe. Together, the data from these experiments suggest that Fe-binding ligands play an important role in the particle exchange occurring in early plumes, and that active production of ligands is a key mechanism for stabilizing a portion of hydrothermal Fe in the far-field plume.

*5.0 Summary and Conclusions*

This study presents four incubation experiments at three separate vent systems along the MAR to document the temporal evolution of physicochemical forms of Fe and Fe-binding ligands to gain insights into the processes effecting Fe early on in hydrothermal plumes. The four incubations captured the range of physical and chemical stages of hydrothermal plume development in the buoyant and neutrally buoyant plumes and the reducing and oxidized plumes.

Exchange between the physicochemical fractions of Fe was observed in each of the incubations with Fe-binding ligands participating in exchange at all stages of plume evolution.

The elevated Fe-binding ligand concentrations observed at the onset of each of the near-field experiments suggests the vent systems along the MAR are a local source of ligands relative to the deep Atlantic. The concentration and binding strength indicate a large concentration of weak ligands characteristic of a pool composed of non-specific organic ligands, produced in serpentinization reactions or by microbial communities at the vent source. These high ligand concentrations declined concurrently with dFe on the timescales of hours via aggregation onto particles, suggesting that most of these sourced Fe-binding ligands may not escape the near-field. The stability of dissolved Fe isotopic composition during this decline in the filtered treatments suggests the complexation and aggregation with Fe was analogous to the flocculation observed with organic matter in estuarine systems. This lends observational support to the low-density organic matrix observed in far-field particles from hydrothermal vents (Fitzsimmons et al., 2017; Hoffman et al., 2020). However, in later stages of these experiments, we observed production of soluble and dissolved Fe-binding ligands all unfiltered treatments as the plumes approached on oxidized state and free Fe(II) was diminished. This promoted particle exchange, driving the enrichment of $\delta^{56}$dFe over a week in the unfiltered treatments TAG and Rainbow near-field experiments. Over longer timescales, ligand production strongly influenced the physicochemical form of Fe, resulting in the solubilization of particles into the colloidal and soluble phases on a timescale of days to weeks in the Lucky Strike and Rainbow far-field experiments. This exchange was associated with a shift in the microbial community from predominantly sulfur and nitrogen oxidizing bacteria to communities dominated by hydrocarbon degraders. We postulate that the transition of hydrothermal plumes from a reducing to oxidized state in its early evolution fosters conditions that promote ligand production from the microbial community and can contribute to the enhanced longevity of hydrothermal Fe in the water column. These experiments suggest that the presence of organic Fe-binding ligands and the formation of inorganic colloids are mutually intertwined in the early evolution of dFe in hydrothermal plume systems.

### 6.0 Acknowledgements
Special thanks to Korinna Kunde for assisting in the preparation of these experiments, David Gonzalez-Santana for initial Fe(II) measurements on subsamples, and Noah Gluschankoff for assistance with DNA collection at sea. Nathan Youlton assisted with DNA extractions. Sharon Walker of the NOAA-Pacific Environmental Laboratory provided the LSS and ORP sensors used for plume identification on this project. We would also like to acknowledge the captain and crew of the RSS *James Cook* who were integral in the collection of the samples for this work. TM was partially supported by the Sanibel-Captiva Shell Club/Mary & Al Bridell Memorial Fellowship. KNB was supported in part by U.S. National Science Foundation awards OCE-1333566 and OCE-2300915. AJM and ML were supported by NE/N010396/1. WW's PhD studentship was funded by the Chinese Scholarship Council and the Graduate School of National Oceanography Center Southampton. DNA sequencing was funded by a University California Santa Barbara Faculty Senate Award to AES. The International GEOTRACES Program is possible in part thanks to the support from the U.S. National Science Foundation (Grant OCE-1840868) and to the Scientific

Committee on Oceanic Research (SCOR). JR was funded by NOAA Ocean Exploration and Earth-
Ocean Interactions programs through the Cooperative Institute for Climate, Ocean, and Ecosystem
Studies; this is CICOES contribution #2023-1273 and PMEL contribution #5519.
***Data availability***
The full dataset is published on Zenodo under DOI 10.5281/zenodo.15214583.
***Author contributions***
TM and KNB designed the experiment. TM carried out the experiments at sea and processed and
analyzed the trace metal and speciation data. AJML, WW, and ML contributed to the setup,
sampling, and analysis and interpretation of Fe isotope data. AES and JBA processed, analyzed,
and aided in interpretation of the 16S rRNA community composition data. PS measured H 2 S
data for select incubations presented in the supplemental data. KNB, AT, ML, JR, and RMB
contributed to the conceptualization and interpretation of the broader study, and supervision of the
preparation of this manuscript.
***Competing interests***
The authors declare that they have no conflict of interest.

***7.0 Work Cited***
Albarede, F. and Beard, B.: Analytical Methods for Non-Traditional Isotopes, Reviews in
Mineralogy and Geochemistry, 55, 113–152, https://doi.org/10.2138/gsrmg.55.1.113, 2004.
Apprill, A., McNally, S., Parsons, R., and Weber, L.: Minor revision to V4 region SSU rRNA
806R gene primer greatly increases detection of SAR11 bacterioplankton, Aquat. Microb. Ecol.,
75, 129–137, https://doi.org/10.3354/ame01753, 2015.
Bennett, S. A., Achterberg, E. P., Connelly, D. P., Statham, P. J., Fones, G. R., and German, C.
R.: The distribution and stabilisation of dissolved Fe in deep-sea hydrothermal plumes, Earth and
Planetary Science Letters, 270, 157–167, https://doi.org/10.1016/j.epsl.2008.01.048, 2008.
Blin, K., Shaw, S., Kautsar, S. A., Medema, M. H., and Weber, T.: The antiSMASH database
version 3: increased taxonomic coverage and new query features for modular enzymes, Nucleic
Acids Research, 49, D639–D643, https://doi.org/10.1093/nar/gkaa978, 2021.
Borer, P. M., Sulzberger, B., Reichard, P., and Kraemer, S. M.: Effect of siderophores on the
light-induced dissolution of colloidal iron(III) (hydr)oxides, Marine Chemistry, 93, 179–193,
https://doi.org/10.1016/j.marchem.2004.08.006, 2005.
Boyle, E. A., Edmond, J. M., and Sholkovitz, E. R.: The mechanism of iron removal in estuaries,
Geochimica et Cosmochimica Acta, 41, 1313–1324, https://doi.org/10.1016/0016-
712 7037(77)90075-8, 1977.

Buck, K. N., Lohan, M. C., Berger, C. J. M., and Bruland, K. W.: Dissolved iron speciation in
two distinct river plumes and an estuary: Implications for riverine iron supply, Limnology &
Oceanography, 52, 843–855, https://doi.org/10.4319/lo.2007.52.2.0843, 2007.
Buck, K. N., Selph, K. E., and Barbeau, K. A.: Iron-binding ligand production and copper
speciation in an incubation experiment of Antarctic Peninsula shelf waters from the Bransfield
Strait, Southern Ocean, Marine Chemistry, 122, 148–159,
https://doi.org/10.1016/j.marchem.2010.06.002, 2010.
Buck, K. N., Moffett, J., Barbeau, K. A., Bundy, R. M., Kondo, Y., and Wu, J.: The organic
complexation of iron and copper: an intercomparison of competitive ligand exchange-adsorptive
cathodic stripping voltammetry (CLE-ACSV) techniques, Limnology & Ocean Methods, 10,
496–515, https://doi.org/10.4319/lom.2012.10.496, 2012.
Buck, K. N., Sohst, B., and Sedwick, P. N.: The organic complexation of dissolved iron along
the U.S. GEOTRACES (GA03) North Atlantic Section, Deep Sea Research Part II: Topical
Studies in Oceanography, 116, 152–165, https://doi.org/10.1016/j.dsr2.2014.11.016, 2015.
Buck, K. N., Gerringa, L. J. A., and Rijkenberg, M. J. A.: An Intercomparison of Dissolved Iron
Speciation at the Bermuda Atlantic Time-series Study (BATS) Site: Results from GEOTRACES
Crossover Station A, Front. Mar. Sci., 3, https://doi.org/10.3389/fmars.2016.00262, 2016.
Buck, K. N., Sedwick, P. N., Sohst, B., and Carlson, C. A.: Organic complexation of iron in the
eastern tropical South Pacific: Results from US GEOTRACES Eastern Pacific Zonal Transect
(GEOTRACES cruise GP16), Marine Chemistry, 201, 229–241,
https://doi.org/10.1016/j.marchem.2017.11.007, 2018.
Bundy, R. M., Abdulla, H. A. N., Hatcher, P. G., Biller, D. V., Buck, K. N., and Barbeau, K. A.:
Iron-binding ligands and humic substances in the San Francisco Bay estuary and estuarine-
influenced shelf regions of coastal California, Marine Chemistry, 173, 183–194,
https://doi.org/10.1016/j.marchem.2014.11.005, 2015.
Bundy, R. M., Boiteau, R. M., McLean, C., Turk-Kubo, K. A., McIlvin, M. R., Saito, M. A., Van
Mooy, B. A. S., and Repeta, D. J.: Distinct Siderophores Contribute to Iron Cycling in the
Mesopelagic at Station ALOHA, Front. Mar. Sci., 5, 61,
https://doi.org/10.3389/fmars.2018.00061, 2018.
Callahan, B. J., McMurdie, P. J., Rosen, M. J., Han, A. W., Johnson, A. J. A., and Holmes, S. P.:
DADA2: High-resolution sample inference from Illumina amplicon data, Nat Methods, 13, 581–
583, https://doi.org/10.1038/nmeth.3869, 2016.
Campbell, A. C., Palmer, M. R., Klinkhammer, G. P., Bowers, T. S., Edmond, J. M., Lawrence,
J. R., Casey, J. F., Thompson, G., Humphris, S., Rona, P., and Karson, J. A.: Chemistry of hot
springs on the Mid-Atlantic Ridge, Nature, 335, 514–519, https://doi.org/10.1038/335514a0,
748  1988.

Conway, T. M. and John, S. G.: Quantification of dissolved iron sources to the North Atlantic
Ocean, Nature, 511, 212–215, https://doi.org/10.1038/nature13482, 2014.
Dideriksen, Baker, J. A., and Stipp, S. L. S.: Fe isotope fractionation between inorganic aqueous
Fe(III) and a Fe siderophore complex, Mineral. mag., 72, 313–316,
https://doi.org/10.1180/minmag.2008.072.1.313, 2008.

Douville, E., Charlou, J. L., Oelkers, E. H., Bienvenu, P., Jove Colon, C. F., Donval, J. P., Fouquet, Y., Prieur, D., and Appriou, P.: The rainbow vent fluids (36°14′N, MAR): the influence of ultramafic rocks and phase separation on trace metal content in Mid-Atlantic Ridge hydrothermal fluids, Chemical Geology, 184, 37–48, https://doi.org/10.1016/S0009-2541(01)00351-5, 2002.

Ellwood, M. J., Strzepek, R. F., Strutton, P. G., Trull, T. W., Fourquez, M., and Boyd, P. W.: Distinct iron cycling in a Southern Ocean eddy, Nat Commun, 11, 825, https://doi.org/10.1038/s41467-020-14464-0, 2020.

Escoube, R., Rouxel, O. J., Sholkovitz, E., and Donard, O. F. X.: Iron isotope systematics in estuaries: The case of North River, Massachusetts (USA), Geochimica et Cosmochimica Acta, 73, 4045–4059, https://doi.org/10.1016/j.gca.2009.04.026, 2009.

Field, M. P. and Sherrell, R. M.: Dissolved and particulate Fe in a hydrothermal plume at 9°45′N, East Pacific Rise:, Geochimica et Cosmochimica Acta, 64, 619–628, https://doi.org/10.1016/S0016-7037(99)00333-6, 2000.

Fishwick, M. P., Sedwick, P. N., Lohan, M. C., Worsfold, P. J., Buck, K. N., Church, T. M., and Ussher, S. J.: The impact of changing surface ocean conditions on the dissolution of aerosol iron, Global Biogeochemical Cycles, 28, 1235–1250, https://doi.org/10.1002/2014GB004921, 2014.

Fitzsimmons, J. N., John, S. G., Marsay, C. M., Hoffman, C. L., Nicholas, S. L., Toner, B. M., German, C. R., and Sherrell, R. M.: Iron persistence in a distal hydrothermal plume supported by dissolved–particulate exchange, Nature Geosci, 10, 195–201, https://doi.org/10.1038/ngeo2900, 2017.

Gartman, A. and Findlay, A. J.: Publisher Correction: Impacts of hydrothermal plume processes on oceanic metal cycles and transport, Nat. Geosci., 13, 654–654, https://doi.org/10.1038/s41561-020-0625-y, 2020.

Gartman, A. and Luther, G. W.: Oxidation of synthesized sub-micron pyrite ($FeS_2$) in seawater, Geochimica et Cosmochimica Acta, 144, 96–108, https://doi.org/10.1016/j.gca.2014.08.022, 2014.

Gartman, A., Findlay, A. J., and Luther, G. W.: Nanoparticulate pyrite and other nanoparticles are a widespread component of hydrothermal vent black smoker emissions, Chemical Geology, 366, 32–41, https://doi.org/10.1016/j.chemgeo.2013.12.013, 2014.

German, C. R., Campbell, A. C., and Edmond, J. M.: Hydrothermal scavenging at the Mid-Atlantic Ridge: Modification of trace element dissolved fluxes, Earth and Planetary Science Letters, 107, 101–114, https://doi.org/10.1016/0012-821X(91)90047-L, 1991.

German, C. R., Lang, S. Q., and Fitzsimmons, J. N.: Marine Hydrothermal processes, in: Treatise on Geochemistry, Elsevier, 145–176, https://doi.org/10.1016/B978-0-323-99762-1.00048-6, 2025.

Gerringa, L. J. A., Rijkenberg, M. J. A., Schoemann, V., Laan, P., and De Baar, H. J. W.:
Organic complexation of iron in the West Atlantic Ocean, Marine Chemistry, 177, 434–446,
https://doi.org/10.1016/j.marchem.2015.04.007, 2015.

Gledhill, M. and Van Den Berg, C. M. G.: Determination of complexation of iron(III) with
natural organic complexing ligands in seawater using cathodic stripping voltammetry, Marine
Chemistry, 47, 41–54, https://doi.org/10.1016/0304-4203(94)90012-4, 1994.

Golyshin, P. N., Chernikova, T. N., Abraham, W.-R., Lünsdorf, H., Timmis, K. N., and
Yakimov, M. M.: Oleiphilaceae fam. nov., to include Oleiphilus messinensis gen. nov., sp. nov.,
a novel marine bacterium that obligately utilizes hydrocarbons., International Journal of
Systematic and Evolutionary Microbiology, 52, 901–911, https://doi.org/10.1099/00207713-52-
3-901, 2002.

González-Santana, D., González-Dávila, M., Lohan, M. C., Artigue, L., Planquette, H., Sarthou,
G., Tagliabue, A., and Santana-Casiano, J. M.: Variability in iron (II) oxidation kinetics across
diverse hydrothermal sites on the northern Mid Atlantic Ridge, Geochimica et Cosmochimica
Acta, 297, 143–157, https://doi.org/10.1016/j.gca.2021.01.013, 2021.

González-Santana, D., Lough, A. J. M., Planquette, H., Sarthou, G., Tagliabue, A., and Lohan,
M. C.: The unaccounted dissolved iron (II) sink: Insights from dFe(II) concentrations in the deep
Atlantic Ocean, Science of The Total Environment, 862, 161179,
https://doi.org/10.1016/j.scitotenv.2022.161179, 2023.

Hassler, C. S., Alasonati, E., Mancuso Nichols, C. A., and Slaveykova, V. I.:
Exopolysaccharides produced by bacteria isolated from the pelagic Southern Ocean — Role in
Fe binding, chemical reactivity, and bioavailability, Marine Chemistry, 123, 88–98,
https://doi.org/10.1016/j.marchem.2010.10.003, 2011.

Hatta, M., Measures, C. I., Wu, J., Roshan, S., Fitzsimmons, J. N., Sedwick, P., and Morton, P.:
An overview of dissolved Fe and Mn distributions during the 2010–2011 U.S. GEOTRACES
north Atlantic cruises: GEOTRACES GA03, Deep Sea Research Part II: Topical Studies in
Oceanography, 116, 117–129, https://doi.org/10.1016/j.dsr2.2014.07.005, 2015.

Hawkes, Gledhill, M., Connelly, D. P., and Achterberg, E. P.: Characterisation of iron binding
ligands in seawater by reverse titration, Analytica Chimica Acta, 766, 53–60,
https://doi.org/10.1016/j.aca.2012.12.048, 2013a.

Hawkes, Connelly, D. P., Gledhill, M., and Achterberg, E. P.: The stabilisation and
transportation of dissolved iron from high temperature hydrothermal vent systems, Earth and
Planetary Science Letters, 375, 280–290, https://doi.org/10.1016/j.epsl.2013.05.047, 2013b.

Hochella, M. F., Lower, S. K., Maurice, P. A., Penn, R. L., Sahai, N., Sparks, D. L., and
Twining, B. S.: Nanominerals, Mineral Nanoparticles, and Earth Systems, Science, 319, 1631–
1635, https://doi.org/10.1126/science.1141134, 2008.

Hoffman, C. L., Schladweiler, C. S., Seaton, N. C. A., Nicholas, S. L., Fitzsimmons, J. N.,
Sherrell, R. M., German, C. R., Lam, P. J., and Toner, B. M.: Diagnostic Morphology and Solid-

State Chemical Speciation of Hydrothermally Derived Particulate Fe in a Long-Range
Dispersing Plume, ACS Earth Space Chem., 4, 1831–1842,
https://doi.org/10.1021/acsearthspacechem.0c00067, 2020.
Hoffman, C. L., Monreal, P. J., Albers, J. B., Lough, A. J. M., Santoro, A. E., Mellett, T., Buck,
K. N., Tagliabue, A., Lohan, M. C., Resing, J. A., and Bundy, R. M.: Microbial strong organic-
ligand production is tightly coupled to iron in hydrothermal plumes, Biogeosciences, 21, 5233–
5246, https://doi.org/10.5194/bg-21-5233-2024, 2024.
Hollister, A. P., Kerr, M., Malki, K., Muhlbach, E., Robert, M., Tilney, C. L., Breitbart, M.,
Hubbard, K. A., and Buck, K. N.: Regeneration of macronutrients and trace metals during
phytoplankton decay: An experimental study, Limnology & Oceanography, 65, 1936–1960,
https://doi.org/10.1002/lno.11429, 2020.
John, S. G. and Adkins, J.: The vertical distribution of iron stable isotopes in the North Atlantic
near Bermuda, Global Biogeochemical Cycles, 26, 2011GB004043,
https://doi.org/10.1029/2011GB004043, 2012.
Johnson, K. S., Gordon, R. M., and Coale, K. H.: What controls dissolved iron concentrations in
the world ocean?, Marine Chemistry, 57, 137–161, https://doi.org/10.1016/S0304-
844 4203(97)00043-1, 1997.

Johnson, K. S., Elrod, V., Fitzwater, S., Plant, J., Boyle, E., Bergquist, B., Bruland, K., Aguilar-
Islas, A., Buck, K., Lohan, M., Smith, G. J., Sohst, B., Coale, K., Gordon, M., Tanner, S.,
Measures, C., Moffett, J., Barbeau, K., King, A., Bowie, A., Chase, Z., Cullen, J., Laan, P.,
Landing, W., Mendez, J., Milne, A., Obata, H., Doi, T., Ossiander, L., Sarthou, G., Sedwick, P.,
Van Den Berg, S., Laglera-Baquer, L., Wu, J., and Cai, Y.: Developing standards for dissolved
iron in seawater, EoS Transactions, 88, 131–132, https://doi.org/10.1029/2007EO110003, 2007.
Kertesz, M. A., Kawasaki, A., and Stolz, A.: Aerobic Hydrocarbon-Degrading
Alphaproteobacteria: Sphingomonadales, in: Taxonomy, Genomics and Ecophysiology of
Hydrocarbon-Degrading Microbes, edited by: McGenity, T. J., Springer International Publishing,
Cham, 105–124, https://doi.org/10.1007/978-3-030-14796-9_9, 2019.
Klar, J. K., James, R. H., Gibbs, D., Lough, A., Parkinson, I., Milton, J. A., Hawkes, J. A., and
Connelly, D. P.: Isotopic signature of dissolved iron delivered to the Southern Ocean from
hydrothermal vents in the East Scotia Sea, Geology, 45, 351–354,
https://doi.org/10.1130/G38432.1, 2017.
Kleint, C., Hawkes, J. A., Sander, S. G., and Koschinsky, A.: Voltammetric Investigation of
Hydrothermal Iron Speciation, Front. Mar. Sci., 3, https://doi.org/10.3389/fmars.2016.00075,
861 2016.

Klunder, M. B., Laan, P., Middag, R., De Baar, H. J. W., and Van Ooijen, J. C.: Dissolved iron
in the Southern Ocean (Atlantic sector), Deep Sea Research Part II: Topical Studies in
Oceanography, 58, 2678–2694, https://doi.org/10.1016/j.dsr2.2010.10.042, 2011.
Kondo, Y., Takeda, S., and Furuya, K.: Distinct trends in dissolved Fe speciation between
shallow and deep waters in the Pacific Ocean, Marine Chemistry, 134–135, 18–28,
https://doi.org/10.1016/j.marchem.2012.03.002, 2012.
Konn, C., Charlou, J. L., Donval, J. P., Holm, N. G., Dehairs, F., and Bouillon, S.: Hydrocarbons
and oxidized organic compounds in hydrothermal fluids from Rainbow and Lost City ultramafic-
hosted vents, Chemical Geology, 258, 299–314, https://doi.org/10.1016/j.chemgeo.2008.10.034,
871    2009.

Lacan, F., Radic, A., Labatut, M., Jeandel, C., Poitrasson, F., Sarthou, G., Pradoux, C., Chmeleff,
J., and Freydier, R.: High-Precision Determination of the Isotopic Composition of Dissolved Iron
in Iron Depleted Seawater by Double Spike Multicollector-ICPMS, Anal. Chem., 82, 7103–
7111, https://doi.org/10.1021/ac1002504, 2010.
Lam, P. J., Bishop, J. K. B., Henning, C. C., Marcus, M. A., Waychunas, G. A., and Fung, I. Y.:
Wintertime phytoplankton bloom in the subarctic Pacific supported by continental margin iron,
Global Biogeochemical Cycles, 20, 2005GB002557, https://doi.org/10.1029/2005GB002557,
879    2006.

Lang, S. Q., Butterfield, D. A., Lilley, M. D., Paul Johnson, H., and Hedges, J. I.: Dissolved
organic carbon in ridge-axis and ridge-flank hydrothermal systems, Geochimica et
Cosmochimica Acta, 70, 3830–3842, https://doi.org/10.1016/j.gca.2006.04.031, 2006.
Li, J., Babcock-Adams, L., Boiteau, R. M., McIlvin, M. R., Manck, L. E., Sieber, M., Lanning,
N. T., Bundy, R. M., Bian, X., Ștreangă, I.-M., Granzow, B. N., Church, M. J., Fitzsimmons, J.
N., John, S. G., Conway, T. M., and Repeta, D. J.: Microbial iron limitation in the ocean's
twilight zone, Nature, 633, 823–827, https://doi.org/10.1038/s41586-024-07905-z, 2024.
Li, M., Toner, B. M., Baker, B. J., Breier, J. A., Sheik, C. S., and Dick, G. J.: Microbial iron
uptake as a mechanism for dispersing iron from deep-sea hydrothermal vents, Nat Commun, 5,
3192, https://doi.org/10.1038/ncomms4192, 2014.
Liu, X. and Millero, F. J.: The solubility of iron in seawater, Marine Chemistry, 77, 43–54,
https://doi.org/10.1016/S0304-4203(01)00074-3, 2002.
Lough, A. J. M., Klar, J. K., Homoky, W. B., Comer-Warner, S. A., Milton, J. A., Connelly, D.
P., James, R. H., and Mills, R. A.: Opposing authigenic controls on the isotopic signature of
dissolved iron in hydrothermal plumes, Geochimica et Cosmochimica Acta, 202, 1–20,
https://doi.org/10.1016/j.gca.2016.12.022, 2017.
Lough, A. J. M., Homoky, W. B., Connelly, D. P., Comer-Warner, S. A., Nakamura, K.,
Abyaneh, M. K., Kaulich, B., and Mills, R. A.: Soluble iron conservation and colloidal iron
dynamics in a hydrothermal plume, Chemical Geology, 511, 225–237,
https://doi.org/10.1016/j.chemgeo.2019.01.001, 2019.
Lough, A. J. M., Tagliabue, A., Demasy, C., Resing, J. A., Mellett, T., Wyatt, N. J., and Lohan,
M. C.: Tracing differences in iron supply to the Mid-Atlantic Ridge valley between hydrothermal
vent sites: implications for the addition of iron to the deep ocean, Biogeosciences, 20, 405–420,
https://doi.org/10.5194/bg-20-405-2023, 2023.
Lupton, J. E., Delaney, J. R., Johnson, H. P., and Tivey, M. K.: Entrainment and vertical
transport of deep-ocean water by buoyant hydrothermal plumes, Nature, 316, 621–623,
https://doi.org/10.1038/316621a0, 1985.
Mahieu, L., Whitby, H., Dulaquais, G., Tilliette, C., Guigue, C., Tedetti, M., Lefevre, D.,
Fourrier, P., Bressac, M., Sarthou, G., Bonnet, S., Guieu, C., and Salaün, P.: Iron-binding by
dissolved organic matter in the Western Tropical South Pacific Ocean (GEOTRACES TONGA
cruise GPpr14), Front. Mar. Sci., 11, 1304118, https://doi.org/10.3389/fmars.2024.1304118,
911     2024.

Manck, L. E., Coale, T. H., Stephens, B. M., Forsch, K. O., Aluwihare, L. I., Dupont, C. L.,
Allen, A. E., and Barbeau, K. A.: Iron limitation of heterotrophic bacteria in the California
Current System tracks relative availability of organic carbon and iron, The ISME Journal, 18,
wrae061, https://doi.org/10.1093/ismejo/wrae061, 2024.
Martin, J. H. and Fitzwater, S. E.: Iron deficiency limits phytoplankton growth in the north-east
Pacific subarctic, Nature, 331, 341–343, https://doi.org/10.1038/331341a0, 1988.
Moore, C. M., Mills, M. M., Arrigo, K. R., Berman-Frank, I., Bopp, L., Boyd, P. W., Galbraith,
E. D., Geider, R. J., Guieu, C., Jaccard, S. L., Jickells, T. D., La Roche, J., Lenton, T. M.,
Mahowald, N. M., Marañón, E., Marinov, I., Moore, J. K., Nakatsuka, T., Oschlies, A., Saito, M.
A., Thingstad, T. F., Tsuda, A., and Ulloa, O.: Processes and patterns of oceanic nutrient
limitation, Nature Geosci, 6, 701–710, https://doi.org/10.1038/ngeo1765, 2013.
Moore, J. K., Doney, S. C., Glover, D. M., and Fung, I. Y.: Iron cycling and nutrient-limitation
patterns in surface waters of the World Ocean, Deep Sea Research Part II: Topical Studies in
Oceanography, 49, 463–507, https://doi.org/10.1016/S0967-0645(01)00109-6, 2001.
Mottl, M. J. and McConachy, T. F.: Chemical processes in buoyant hydrothermal plumes on the
East Pacific Rise near 21°N, Geochimica et Cosmochimica Acta, 54, 1911–1927,
https://doi.org/10.1016/0016-7037(90)90261-I, 1990.
Nishioka, J., Obata, H., and Tsumune, D.: Evidence of an extensive spread of hydrothermal
dissolved iron in the Indian Ocean, Earth and Planetary Science Letters, 361, 26–33,
https://doi.org/10.1016/j.epsl.2012.11.040, 2013.
Omanović, D., Garnier, C., and Pižeta, I.: ProMCC: An all-in-one tool for trace metal
complexation studies, Marine Chemistry, 173, 25–39,
https://doi.org/10.1016/j.marchem.2014.10.011, 2015.
Parada, A. E., Needham, D. M., and Fuhrman, J. A.: Every base matters: assessing small subunit
rRNA primers for marine microbiomes with mock communities, time series and global field
samples, Environmental Microbiology, 18, 1403–1414, https://doi.org/10.1111/1462-
938     2920.13023, 2016.

Pester, N. J., Reeves, E. P., Rough, M. E., Ding, K., Seewald, J. S., and Seyfried, W. E.:
Subseafloor phase equilibria in high-temperature hydrothermal fluids of the Lucky Strike
Seamount (Mid-Atlantic Ridge, 37°17′N), Geochimica et Cosmochimica Acta, 90, 303–322,
https://doi.org/10.1016/j.gca.2012.05.018, 2012.
Pižeta, I., Sander, S. G., Hudson, R. J. M., Omanović, D., Baars, O., Barbeau, K. A., Buck, K.
N., Bundy, R. M., Carrasco, G., Croot, P. L., Garnier, C., Gerringa, L. J. A., Gledhill, M., Hirose,
K., Kondo, Y., Laglera, L. M., Nuester, J., Rijkenberg, M. J. A., Takeda, S., Twining, B. S., and
Wells, M.: Interpretation of complexometric titration data: An intercomparison of methods for
estimating models of trace metal complexation by natural organic ligands, Marine Chemistry,
173, 3–24, https://doi.org/10.1016/j.marchem.2015.03.006, 2015.
Quast, C., Pruesse, E., Yilmaz, P., Gerken, J., Schweer, T., Yarza, P., Peplies, J., and Glöckner,
F. O.: The SILVA ribosomal RNA gene database project: improved data processing and web-
based tools, Nucleic Acids Research, 41, D590–D596, https://doi.org/10.1093/nar/gks1219,
952  2012.

Resing, J. A., Sedwick, P. N., German, C. R., Jenkins, W. J., Moffett, J. W., Sohst, B. M., and
Tagliabue, A.: Basin-scale transport of hydrothermal dissolved metals across the South Pacific
Ocean, Nature, 523, 200–203, https://doi.org/10.1038/nature14577, 2015.
Revels, B. N., Ohnemus, D. C., Lam, P. J., Conway, T. M., and John, S. G.: The isotopic
signature and distribution of particulate iron in the North Atlantic Ocean, Deep Sea Research
Part II: Topical Studies in Oceanography, 116, 321–331,
https://doi.org/10.1016/j.dsr2.2014.12.004, 2015.
Rudnicki, M. D. and Elderfield, H.: A chemical model of the buoyant and neutrally buoyant
plume above the TAG vent field, 26 degrees N, Mid-Atlantic Ridge, Geochimica et
Cosmochimica Acta, 57, 2939–2957, https://doi.org/10.1016/0016-7037(93)90285-5, 1993.
Rue, E. L. and Bruland, K. W.: Complexation of iron(III) by natural organic ligands in the
Central North Pacific as determined by a new competitive ligand equilibration/adsorptive
cathodic stripping voltammetric method, Marine Chemistry, 50, 117–138,
https://doi.org/10.1016/0304-4203(95)00031-L, 1995.
Sander, S. G. and Koschinsky, A.: Metal flux from hydrothermal vents increased by organic
complexation, Nature Geosci, 4, 145–150, https://doi.org/10.1038/ngeo1088, 2011.
Santoro, A. E., Casciotti, K. L., and Francis, C. A.: Activity, abundance and diversity of
nitrifying archaea and bacteria in the central California Current, Environmental Microbiology,
12, 1989–2006, https://doi.org/10.1111/j.1462-2920.2010.02205.x, 2010.
Severmann, S., Johnson, C. M., Beard, B. L., German, C. R., Edmonds, H. N., Chiba, H., and
Green, D. R. H.: The effect of plume processes on the Fe isotope composition of hydrothermally
derived Fe in the deep ocean as inferred from the Rainbow vent site, Mid-Atlantic Ridge,
36°14′N, Earth and Planetary Science Letters, 225, 63–76,
https://doi.org/10.1016/j.epsl.2004.06.001, 2004.

Sieber, M., Conway, T. M., De Souza, G. F., Hassler, C. S., Ellwood, M. J., and Vance, D.: Isotopic fingerprinting of biogeochemical processes and iron sources in the iron-limited surface Southern Ocean, Earth and Planetary Science Letters, 567, 116967, https://doi.org/10.1016/j.epsl.2021.116967, 2021.

Skulan, J. L., Beard, B. L., and Johnson, C. M.: Kinetic and equilibrium Fe isotope fractionation between aqueous Fe(III) and hematite, Geochimica et Cosmochimica Acta, 66, 2995–3015, https://doi.org/10.1016/S0016-7037(02)00902-X, 2002.

Stephens, B. M., Opalk, K., Petras, D., Liu, S., Comstock, J., Aluwihare, L. I., Hansell, D. A., and Carlson, C. A.: Organic Matter Composition at Ocean Station Papa Affects Its Bioavailability, Bacterioplankton Growth Efficiency and the Responding Taxa, Front. Mar. Sci., 7, 590273, https://doi.org/10.3389/fmars.2020.590273, 2020.

Sylvan, J. B., Pyenson, B. C., Rouxel, O., German, C. R., and Edwards, K. J.: Time-series analysis of two hydrothermal plumes at 9°50′N East Pacific Rise reveals distinct, heterogeneous bacterial populations, Geobiology, 10, 178–192, https://doi.org/10.1111/j.1472-4669.2011.00315.x, 2012.

Tagliabue, A., Aumont, O., and Bopp, L.: The impact of different external sources of iron on the global carbon cycle, Geophysical Research Letters, 41, 920–926, https://doi.org/10.1002/2013GL059059, 2014.

Tivey, M.: Generation of Seafloor Hydrothermal Vent Fluids and Associated Mineral Deposits, Oceanog., 20, 50–65, https://doi.org/10.5670/oceanog.2007.80, 2007.

Tortell, P. D., Maldonado, M. T., Granger, J., and Price, N. M.: Marine bacteria and biogeochemical cycling of iron in the oceans, FEMS Microbiology Ecology, 29, 1–11, https://doi.org/10.1111/j.1574-6941.1999.tb00593.x, 1999.

Vraspir, J. M. and Butler, A.: Chemistry of Marine Ligands and Siderophores, Annu. Rev. Mar. Sci., 1, 43–63, https://doi.org/10.1146/annurev.marine.010908.163712, 2009.

Wang, W., Lough, A., Lohan, M. C., Connelly, D. P., Cooper, M., Milton, J. A., Chavagnac, V., Castillo, A., and James, R. H.: Behavior of iron isotopes in hydrothermal systems: Beebe and Von Damm vent fields on the Mid-Cayman ultraslow-spreading ridge, Earth and Planetary Science Letters, 575, 117200, https://doi.org/10.1016/j.epsl.2021.117200, 2021.

Welch, S. A., Beard, B. L., Johnson, C. M., and Braterman, P. S.: Kinetic and equilibrium Fe isotope fractionation between aqueous Fe(II) and Fe(III), Geochimica et Cosmochimica Acta, 67, 4231–4250, https://doi.org/10.1016/S0016-7037(03)00266-7, 2003.

Wiederhold, J. G., Kraemer, S. M., Teutsch, N., Borer, P. M., Halliday, A. N., and Kretzschmar, R.: Iron Isotope Fractionation during Proton-Promoted, Ligand-Controlled, and Reductive Dissolution of Goethite, Environ. Sci. Technol., 40, 3787–3793, https://doi.org/10.1021/es052228y, 2006.

Wu, J., Wells, M. L., and Rember, R.: Dissolved iron anomaly in the deep tropical–subtropical
Pacific: Evidence for long-range transport of hydrothermal iron, Geochimica et Cosmochimica
Acta, 75, 460–468, https://doi.org/10.1016/j.gca.2010.10.024, 2011.
Yakimov, M. M., Golyshin, P. N., Lang, S., Moore, E. R. B., Abraham, W.-R., Lunsdorf, H., and
Timmis, K. N.: Alcanivorax borkumensis gen. nov., sp. nov., a new, hydrocarbon-degrading and
surfactant-producing marine bacterium, International Journal of Systematic Bacteriology, 48,
339–348, https://doi.org/10.1099/00207713-48-2-339, 1998.
Yücel, M., Gartman, A., Chan, C. S., and Luther, G. W.: Hydrothermal vents as a kinetically
stable source of iron-sulphide-bearing nanoparticles to the ocean, Nature Geosci, 4, 367–371,
https://doi.org/10.1038/ngeo1148, 2011.
Zhang, R., John, S. G., Zhang, J., Ren, J., Wu, Y., Zhu, Z., Liu, S., Zhu, X., Marsay, C. M., and
Wenger, F.: Transport and reaction of iron and iron stable isotopes in glacial meltwaters on
Svalbard near Kongsfjorden: From rivers to estuary to ocean, Earth and Planetary Science
Letters, 424, 201–211, https://doi.org/10.1016/j.epsl.2015.05.031, 2015.
Zykwinska, A., Marchand, L., Bonnetot, S., Sinquin, C., Colliec-Jouault, S., and Delbarre-
Ladrat, C.: Deep-sea Hydrothermal Vent Bacteria as a Source of Glycosaminoglycan-Mimetic
Exopolysaccharides, Molecules, 24, 1703, https://doi.org/10.3390/molecules24091703, 2019.