# Peer review of "Organic iron-binding ligands mediate dissolved-particulate exchange in hydrothermal vent plumes along the mid-Atlantic Ridge"

_EGUsphere, 2025_

## Author Response (AR1)

**Reviewer #1**

**Dear Authors,**

This is a novel and insightful study that contributes to our understanding of the biogeochemical cycle of iron, particularly in hydrothermal regions. The manuscript will be supported for publication once the following issues are addressed. I would like to thank the authors this effort to have all the amount of analysis for the paper from that cruise.

*We thank the reviewers for their support and for their helpful comments. We have attempted to address each issue raised, and have responded to each comment in italics below.*

**Minor Comments:**

- Please correct the term "physiochemical" to "physico-chemical" throughout the manuscript.

*Corrected throughout, thank you.*

- Lines 97–102: This information may be omitted from the introduction as it is repeated in the experimental section.

*Omitted.*

- The introduction should include an overview of existing incubation studies found in the literature.

*Thank you for this suggestion. We have added an overview of Lough et al. 2017, which is to our knowledge the only other incubation study of hydrothermal iron.*

- Lines 196–198: This sentence is redundant with previous content and could be removed.

*Removed.*

**Technical Issues:**

- The authors should justify the use of two different artificial ligands for LFe determination. Since both reverse and forward titration methods can use the same

ligand, this would allow for direct comparison of conditional stability constants, which are ligand-dependent.

*Although both ligands could indeed be used for forward and reverse titrations, we chose to use what we thought would be the optimal approach for each. SA has been vetted in intercalibration exercises, widely applied to forward titrations of Fe speciation in seawater, and forward titration results with SA have been incorporated in the GEOTRACES Data Products, allowing synthesis and direct comparison of our experimental work with water column results also achieved through forward titrations. Similarly, NN has been well characterized for reverse titrations of Fe speciation and applied previously to reverse titrations of hydrothermal plume samples, maintaining some consistency between our experimental results here and previously conducted field work with this method. We have added text in the methods section to incorporate these justifications.*

Cheize et al. (2012) demonstrated that NN is suitable for low-salinity solutions. However, in this incubation study, samples could be diluted using UV-irradiated seawater or NaCl to maintain constant ionic strength. Altering ionic strength significantly influences physico-chemical processes and iron speciation.

Cheize, M., et al. (2012). Iron organic speciation determination in rainwater using cathodic stripping voltammetry. Analytica Chimica Acta, 736, 45–54.

*Here we applied the NN method following the recommendation of Hawkes et. at. 2013 for hydrothermal samples, which used ultrapure water for the sample dilutions. The Hawkes 2013 study did test different dilution factors (20x, 10x, and 5x), and found the salinity difference to have no effect on the $K_{FeL}$ values, though lower salinities did cause an overestimate of L concentrations. We used the lower dilution factor of 5x and have noted this possible overestimation in the methods section (lines 245–247).*

- Please justify the selected detection window with SA and explain why an alternative was not used that might allow the identification of weaker ligands.

*Of course - we were especially interested in the role of strong ligands on mediating physico-chemical exchange of Fe in hydrothermal plumes, and this detection window has been successfully applied to characterizing stronger model iron-binding ligands. As noted above, we also chose this method with SA to allow synthesis with GEOTRACES field studies. We have added this justification to the text as well (Lines 211–213).*

- Why is the equilibration time with SA limited to only one hour? Justification is needed, as most studies. including those by van den Berg, typically use overnight equilibration.

*The equilibration with SA is quite fast when added after the Fe additions in the titration (Rue and Bruland 1995), so although overnight equilibration times can be applied, they are not necessary (Mahieu et al. 2024). Both shorter and overnight SA equilibrations provide the same conditional stability constants for the SA determined from competition with EDTA (Rue and Bruland 1995; Buck et al. 2007; Abualhaija and van den Berg 2014), and the same L and K results for samples (Mahieu et al. 2024).*

- What is the concentration of Fe(II) in the samples? This is important because the voltammetric method used detects only Fe(III). If Fe(II) is present in significant and stable amounts, LFe values could be overestimated.

*There was some Fe(II) present at the start of all the reducing plume incubations (González-Santana et al. 2023). Initial Fe(II) concentrations were ~0.8 nM at Lucky Strike, ~70 nM at Rainbow near-field, and ~10 nM at TAG. We have added these details to the results section for each near-field experiment. However, by the time the samples were frozen, thawed, equilibrated, and analyzed back at the University of South Florida weeks later, we expect that any Fe(II) would be completely oxidized to Fe(III).*

*González-Santana, D., Lough, A. J., Planquette, H., Sarthou, G., Tagliabue, A., & Lohan, M. C. (2023). The unaccounted dissolved iron (II) sink: Insights from dFe (II) concentrations in the deep Atlantic Ocean. Science of the Total Environment, 862, 161179.*

- In the forward titrations, are the samples diluted? If so, was salinity corrected accordingly?

*No dilutions were conducted for any forward titrations. We have clarified this in the Methods sections (lines 247). We have also indicated the dilutions in the data tables in supplementary information.*

- How did the pH evolve during the incubations? pH plays a key role in Fe(II) oxidation and associated physico-chemical processes.

*Potentiometric pH measurements were taken along this cruise and some opportunistic measurements were made in a few of these incubations. We didn't observe any large changes in pH, with the largest difference being ~0.1 units from the environmental samples (Gonzalez-Santana et al., 2023, reference provided above). That said, we also buffered all titrations to pH 8.2 prior to competition with the competitive ligand so our results are all from the same pH conditions.*

- In Figures 2c and 2e, dFe (and consequently dLFe) concentrations increase on Day 1. How is this explained? Were replicate measurements performed?

*We found this unusual as well, and did make replicate measurements of this sample. The total Fe within this incubation was extremely high (>5,000 nM), with an increase in soluble Fe as well as $H_2S$ concentrations (Supplemental Figure 7a) just before day 1 (sampling was limited for Fe(II) and $H_2S$ as field samples were prioritized for these time sensitive measurements). Thus, we believe the day 1 increase most likely reflects initial dissociation of pyrite particles originally formed at the vent-ocean interface followed by the rapid formation of colloids (presumably Fe(oxy)-hydroxides) observed in the 24 hour time point before aggregating into the particulate phase. These changes in the physicochemical speciation likely impacted lability of dFe that could be liberated by the added ligand NN in this sample, even if reduced species were not present at the time of analysis. This may have increased the maximum recovered from NN and thus the modeled L concentration from this sample.*

- The manuscript refers to differences in logK values; however, these do not appear to be significant in the figures. Please provide a statistical analysis to determine whether the variations are significant.

*Thank you for this suggestion. We performed a t-test to test the significance of differences highlighted in the text and to add context to these statements. These statistics have now been added to the manuscript. For some experiments there was a significant difference between soluble and dissolve phase ligand log K values but in others there were no statistical significance. We have added statements on Lines 349, 426–428, 470–471, and 511–513.*

- In Figure 5b, how do the authors explain that sFe concentrations exceed cFe concentrations?

*This likely reflects analytical uncertainty in the measurements, with this sample having a dFe concentration of $6.78 \pm 0.52$ and a sFe concentration of $7.95 \pm 0.53$. We interpret these results to reflect 100% of the dissolved Fe in the soluble phase, and the separation driven either by a slight underestimation of dFe or overestimation of sFe.*

I hope this constructive revision will help the authors to improve the manuscript.

*Yes, thank you!*

Regards
**Citation**: https://doi.org/10.5194/egusphere-2025-1798-RC1

**Reviewer #2**

Mellett et al studied the temporal evolution of Fe speciation and fractionation at Atlantic HT plumes using incubation experiments. The study very neatly shows physicochemical evolution of Fe and ligands - their initial introduction in the near systems, quick floccolation / scavenging and particularly the downrange introduction of complexed Fe.

The manuscript is a pleasant read, with little to remark on layout and typography.

*We thank the reviewer for their supportive feedback and helpful comments on the manuscript. We have addressed each of the comments and provided our responses in italics below.*

This reviewers main concern is where part of the discussion posits strong ligands, particularly siderophores, having a role in the down-plume stabilisation of Fe. This part of the discussion is hard to reconcile with one of the main findings of the incubation experiments being an overall weaker ligand pool. These sections (towards the ends of both sections 4.1 and 4.2), require better justification. In the opinion of this reviewer the identification of families known for *putative* siderophore pathways in the face of an overall weaker ligand pool needs more convincing arguments for a meaningful role of siderophores in the speciation in HT systems.

*Thank you for identifying this confusion. We have revised the discussion sections 4.1 and 4.2 to more clearly articulate our interpretation of the results as highlighting the presence of two distinct ligand pools, one present initially near the vent itself (the weaker ligand pool) and one that is produced in-situ by the microbial community over time (the stronger ligand pool). Our datasets, which are coherent with the field-based studies, suggest that both this initial weak ligand pool at the vent and the emergence of stronger, siderophore-like ligands in the plume are important for the stabilization and transport of dFe down-plume.*

*We absolutely acknowledge that without siderophore samples from these specific experiments, we are relying on more circumstantial, though we think compelling, evidence to support the interpretation of siderophores contributing to a portion of dFe stabilization within plume systems:*

*1. Siderophores were actually identified within the plumes in field samples taken from the same cruise (Hoffman et al., 2024).*

*2. Siderophores would fall within the soluble fraction of the ligand pool, the ligand pool that increased at later stages of many of the unfiltered incubations. Siderophores are also somewhat uniquely known to solubilize Fe from larger size fractions, and the increase in soluble stronger ligands we observed in the incubations was accompanied by an increase also in the soluble Fe fraction.*

*3. The heavier fractionation of dFe isotopes in the later stages of unfiltered treatments at TAG and Rainbow near-field were also consistent with increasing organic complexation by siderophore-like ligands.*

*4. 16S data showed that the microbial community at the Family level contained putative siderophore pathways, and these microbial groups increased in abundance towards the end of many experiments.*

*We think that altogether these lines of evidence support the hypothesis of siderophore production over time in the experiments, and we have edited the text to more clearly show our logic and to highlight the uncertainties in these observations. See lines 587–603 in section 4.1 and further context to the isotopic fractionation in final two paragraphs of section 4.2.*

specific remarks

physiochemical used throughout the manuscript, authors probably mean to say physicochemical.

*Thank you for catching this. This has been changed throughout the manuscript.*

figures 2-5: a lot is happening in the first days, suggest making the horizontal axes nonlinear to better show the quick successions taking place initially.

*Thank you for this suggestion. We did try plotting these figures using log-scale x-axis. The transformation was not useful on longer incubations (Figures 4 and 5) and ended up compressing these datasets, but did provide some benefits to spreading out the condensed early sampling in the short-term incubations (Figure 2 and Figure 3). However, in Figure 3 this visualization ended up compressing data on the other end of sampling time. We did notice that the log plots appeared very similar to physiochemical breakdown in the figures (a, b) due to the even spacing of the bar plots. Thus, we ultimately decided to maintain consistency amongst all plots for simplicity, and hope that the physicochemical components in the first panel captures the quick succession of Fe in early sampling experiments. We have changed the scale of the y-axes in Figure 2a to more clearly illustrate the changes observed. Thank you for the thoughtful suggestion.*

587-589 Please provide additional justification for the link between Fe-limited Southern Ocean surface water processes to the present findings.

*This sentence was attempting to draw comparisons between the dFe isotope data in this study to others that point to amplified dFe isotopic signatures in systems with high cycling of Fe. We have edited this text to add more context and an additional source to Lines 780–784.*

590-592 Please provide additional justification of Hoffman et al. (2024) finding strong ligands in HT systems relating to the present findings.

*Hoffman et al. (2024) was able to quantify siderophores in water column samples from the same cruise, at the same locations as the experiments that were done in this study. Although we did not have sufficient water volume to allow the same approach for measuring siderophores in these experiments, we think there is strong evidence that the increase in soluble organic ligands we measured in this study could be due to the production of siderophores. We have edited this section of the discussion to more clearly tie these studies together and to outline our train of thought.*

*Thank you for all of your helpful suggestions, we really appreciate your efforts to help improve this manuscript!*

**Citation**: https://doi.org/10.5194/egusphere-2025-1798-RC2